# Fixed Budget Best Arm Identification in Unimodal Bandits

**Debamita Ghosh** *debamita.ghosh@iitb.ac.in*
*IITB-Monash Research Academy*
*Indian Institute of Technology Bombay, India*

**Manjesh K. Hanawal** *mhanawal@iitb.ac.in*
*Department of Industrial Engineering & Operations Research*
*Indian Institute of Technology Bombay, India*

**Nikola Zlatanov** *n.zlatanov@innopolis.ru*
*Faculty of Computer and Engineering Sciences*
*Innopolis University, Russia*

**Reviewed on OpenReview:** *https: // openreview. net/ forum? id= epcLNhkoEL&noteId= g5BMyjdTBG*

## Abstract

We consider the best arm identification problem in a fixed budget stochastic multi-armed bandit in which arm means exhibit unimodal structure, i.e., there is only one local maximum. We establish that the probability of misidentifying the optimal arm within a budget of $T$ is lower bounded as $\mathcal{O}\left(\exp\left\{-T/\bar{H}\right\}\right)$, where $\bar{H}$ depends on the sub-optimality gaps of arms in the neighborhood of the optimal arm. In contrast to the lower bound for the unstructured case, the error exponent in this bound does not depend on the number of arms $K$ and is smaller by a factor $\log K$, which captures the gain achievable by exploiting the unimodal structure. We then develop an algorithm named *Fixed Budget Best Arm Unimodal Bandits (FB-BAUB)* that exploits unimodality to achieve the gain. Specifically, we show that the error probability of FB-BAUB is upper bounded as $\mathcal{O}\left(\log_2 K \exp\left\{-T\Delta^2\right\}\right)$, where $\Delta$ is the gap between the neighboring arms and $\bar{H} \leq 2\Delta^{-2}$. We demonstrate that FB-BAUB outperforms the state-of-the-art algorithms through extensive simulations. Moreover, FB-BAUB is parameter-free and simple to implement.

## 1 Introduction

Multi-armed bandit (MAB) is a popular setup to study decision-making under uncertainty. It has been applied in drug trials, recommendation systems, auctions, communication networks, and the list is growing, see Bouneffouf & Rish (2019). In MAB, a policy is any strategy that sequentially selects arms based on past observations. The performance of a policy is evaluated using criteria such as cumulative regret, simple regret, or best arm identification (BAI), depending on the application, refer to Lattimore & Szepesvári (2020) for details. While the policies optimizing cumulative regret balance exploration and exploitation optimally, the policies optimizing simple regret, or BAI, focus on optimal exploration. In BAI, the goal is to minimize the sample complexity of identifying the best arm within a given tolerance on the error probability (fixed confidence setting) or to minimize the error probability of not identifying the best arm within a given number of rounds (fixed budget setting), see (Karnin et al., 2013; Carpentier & Locatelli, 2016; Atsidakou et al., 2022). We focus on the fixed budget BAI setting.

The classical MAB setup considers the unstructured case, where reward distributions of the arms are independent with no relation among their means. However, many practical problems exhibit structural properties such as smoothness, linearity, unimodality, and convexity on the mean rewards, which can be exploited to improve the performance of the MAB algorithms (Magureanu et al., 2014; Combes et al., 2017; Yu & Mannor, 2011; Cheshire et al., 2021). Our objective in this paper is to develop algorithms that exploit the unimodal structure of the arm means to improve learning performance.

In unimodality, the arms' means are increasing and then decreasing in the arms' indices, exhibiting a change in the monotonicity pattern only at the globally optimal arm. Many real-life applications of unimodal structure arise in network throughput, see Hashemi et al. (2018), sequential pricing, and bidding in online sponsored search auctions, see Yu & Mannor (2011). Unimodality is exploited in Combes & Proutiere (2014); Saber et al. (2020a) to improve the cumulative regret bounds, where it is shown that the regret bound is asymptotically optimal and does not depend on the number of arms. Unimodlity structure is also exploited in Blinn et al. (2021), where the transmitter identifies the best beam in the fixed confidence setting, which is aligned with the receiver's beam in the wireless network. However, in wireless networks where devices (e.g., base stations and mobiles) need to operate synchronously, fixed budget BAI is better suited as the devices can know exactly when to switch from exploration to exploitation and hence require less information exchange for synchronization. This simplifies the protocol design. To the best of our knowledge, the fixed budget BAI with unimodal structure has not been studied in the literature. Also, as noted in Atsidakou et al. (2022), fixed budget BAI is more challenging than their fixed confidence counterparts, and our work fills this gap.

We develop an algorithm named *Fixed Budget Best Arm in Unimodal Bandits (FB-BAUB)* to address the BAI problem with the unimodal structure. The algorithm is motivated by the *Line Search Eliminations (LSE)* algorithm, introduced in Yu & Mannor (2011) and works in phases. In each phase of FB-BAUB, a portion of the arms is eliminated ($\sim 33\%$) based on empirical means of the arms, thereby reducing the search space for the optimal arm in the subsequent phases. For a given budget $T$ and the number of arms $K$, we show that FB-BAUB achieves an error probability of the order of $\mathcal{O}(\log K \exp(-T\Delta^2))$ under the assumption of the minimum separation gap between neighboring arms is at least $\Delta > 0$. Under the same minimum gap assumption, the best known achievable error probability (by Sequential Halving) is of order $\mathcal{O}\left(\log K \exp\left(-\frac{T\Delta^2}{\log K}\right)\right)$ for the unstructured case, see Audibert et al. (2010). Thus, by exploiting the unimodal structure, we reduced the error exponent by factor $\log K$. We show that this reduction is the best possible by establishing a lower bound. Thus, we quantify the gain one can achieve by exploiting the unimodal structure. In establishing the lower bound, we generalize the "flipping constructions" of the bandit instance from Carpentier & Locatelli (2016) to include distribution with unbounded support and adapt it to unimodal bandits.

In summary, our contributions are as follows:

- We establish a lower bound for the unimodal bandits in a fixed budget BAI setting.

- We develop a parameter-free algorithm, named FB-BAUB, and derived an upper bound on its error probability. We show that the error exponent in the bound does not depend on $K$. The lower bound shows that the error exponent of FB-BAUB is of the right order.

- We empirically validate the superior performance of FB-BAUB compared to the other state-of-the-art algorithms in the literature that exploit the unimodal structure.

Detailed proofs of all the statements are given in Appendix A.

## 2 Related Work

The BAI problem is well-studied in the unstructured bandits in both fixed confidence and fixed budget settings, see Mannor & Tsitsiklis (2004); Even-Dar et al. (2006); Kalyanakrishnan et al. (2012); Karnin et al. (2013); Kaufmann et al. (2016); Jamieson & Nowak (2014); Garivier & Kaufmann (2016); Atsidakou et al. (2022); Wang et al. (2021). The authors in Gabillon et al. (2012) develop a unifying approach to analyse both settings leading to a meta-algorithm that can be applied to both settings. The lower bounds for these settings for unstructured bandits are developed by Chen & Li (2015); Carpentier & Locatelli (2016); Audibert et al. (2010).

**Improving Cumulative Regret:** Several works exploit the structural properties of the bandits to minimize cumulative regret. Abbasi-Yadkori et al. (2011); Chu et al. (2011); Dani et al. (2008) exploit *linearity* of observed rewards. The authors in Cesa-Bianchi & Lugosi (2012); Combes et al. (2015) studied *Combinatorial* bandits with bandit feedback exploit the *combinatorial* structure of the arms to learn the best subset of arms.

*Unimodal* structure is exploited in Yu & Mannor (2011); Combes & Proutiere (2014) to improve the regret performance. Combes & Proutiere (2014) provide a lower bound on any policy exploiting unimodal structure and develop an Optimal Sampling for Unimodal Bandits (OSUB) algorithm with a matching upper bound. Saber et al. (2020b) developed algorithms that do not require force explorations as used in the OSUB. The *smoothness* of the rewards expressed in terms of *Lipschitz* conditions are studied by Magureanu et al. (2014); Valko et al. (2014); Hanawal et al. (2015). A generic framework for analysing the cumulative regret of bandits exhibiting structural properties that include linearity, smoothness, and unimodality property is given in Combes et al. (2017). All of the aforementioned works are in the cumulative regret minimization setting.

**Improving Best Arm Identification:** In the best arm identification setting, the authors in Soare et al. (2014); Jedra & Proutiere (2020); Azizi et al. (2022) exploit the linearity structures, and the authors in Kocák & Garivier (2020) exploit the smoothness structures to improve performance of BAI algorithms. The authors in Wang et al. (2021) studied fixed confidence BAI problem and developed Frank-Wolfe-based Sampling (FWS) whose sample complexity matches the lower bounds for a wide class of pure exploration problems. They applied FWS to other structural bandits such as threshold, linear, and Lipschitz, but they did not consider unimodal bandits. The authors in Garivier et al. (2017) consider threshold bandit problems (TBP), where the means of the arms are monotonically increasing, and the goal is to identify the set of arms with means above a given threshold. The authors characterise the sample complexity of the TBP in the fixed confidence setting. The TBP is extended to include other structural properties such as unimodality and concavity in Cheshire et al. (2020), and problem-independent bounds on the simple regret are derived. The paper Cheshire et al. (2021) extended the analysis of the TBP with monotonicity and concavity to establish problem-dependent bounds.

**Lower Bounds:** The lower bounds for the best arm identification are explored in Audibert et al. (2010); Garivier & Kaufmann (2016), and Carpentier & Locatelli (2016). In the fixed budget setting, Audibert et al. (2010) established a first lower bound, and it was improved in Garivier & Kaufmann (2016) using 'flipping constructions'. Notably, these approaches assume a parametric form (Bernoulli and Gaussian) for the underlying distributions and establish a minimax bound. Further, refining the flipping argument, Carpentier & Locatelli (2016) derived a lower bound that matches with the upper bound of the Successive Reject algorithm introduced in Audibert et al. (2010), thus establishing a tighter lower bound. The case of non-parametric bandits is studied in Barrier et al. (2023), which concentrates on developing instance-independent bounds for fixed budget scenarios. However, these bounds hold only asymptotically. The above work deals with the unstructured case, whereas we deal with the structured (unimodal) bandits.

Our work is closer to Cheshire et al. (2020), Yu & Mannor (2011), and Carpentier & Locatelli (2016). Cheshire et al. (2020) focuses on TBPs to exploit the unimodal property and provide a guarantee on the simple regret, which is different from our setting. The LSE algorithm introduced in Yu & Mannor (2011) provides the PAC guarantees in the fixed confidence setting for a continuous set of arms. The algorithm runs in phases, and the arms played in each phase are selected based on the golden ratio. Though our algorithm adapts the basic ideas of LSE, it differs in how arms are selected and eliminated. Also, it does not require any problem-dependent information. The details are given in Subsection 5.1.

## 3 Problem Setup

### 3.1 Notations

We consider the stochastic multi-armed bandit setting where the learner explores a finite set $\mathcal{A} = \{1, 2, \ldots, K\}$ of arms over a fixed horizon $T$. The reward sequence for arm $k \in \mathcal{A}$ corresponds to independently and identically (i.i.d.) samples drawn from a distribution $p(\mu_k)$ with an unknown mean $\mu_k$, i.e., $\mu_k = \mathbb{E}_{X \sim p(\mu_k)}[X]$. We assume $p(\mu_k)$ is $\beta$ sub-Gaussian, where $\beta > 0$, $\forall k \in \mathcal{A}$.

Let $\epsilon_\beta$ denote the set of all bandits instances that are $\beta$ sub-Gaussian with distinct arm means. For any instance $\boldsymbol{p}(\boldsymbol{\mu}) := \{p(\mu_1), \ldots, p(\mu_K)\} \in \epsilon_\beta$ with mean rewards of the arms as $\boldsymbol{\mu} := \{\mu_1, \ldots, \mu_K\}$, let $k^* := k^*\big(\boldsymbol{p}(\boldsymbol{\mu})\big) = \arg\max_{k \in \mathcal{A}} \mu_k$ denote the optimal arm, i.e., the arm with the highest mean. We denote $\epsilon_U$ as the set of bandits in $\epsilon_\beta$ satisfying unimodality, defined below.

**Definition 1.** *(Unimodality):   A bandit instance $\boldsymbol{p}(\boldsymbol{\mu}) \in \epsilon_\beta$ is said to be unimodal iff $\mu_1 < \mu_2 < \cdots < \mu_{k^*}$ and $\mu_{k^*} > \mu_{k^*+1} > \cdots > \mu_K$.*

Let $\mathcal{S}$ be the set of the arm means satisfies the unimodal structure, i.e.,

$$\mathcal{S} = \big\{ \boldsymbol{\mu} : \mu_1 < \mu_2 < \cdots < \mu_{k^*} > \mu_{k^*+1} > \cdots > \mu_K \big\}.$$

### 3.2 Learning setup

We consider the following interaction between the learner and the environment over fixed rounds $T > 0$. We refer to $T$ as the fixed budget. For any round $1 \leq t \leq T$, the learner chooses an arm $k_t \in \mathcal{A}$ and observes a stochastic reward drawn from the distribution $p(\mu_{k_t})$. In each round $t$, the learner decides which arm to play based on the samples observed in the past. At the end of $T$, the learner returns an arm $k_T \in \mathcal{A}$. A policy $\pi$ is any strategy that selects an arm in each round, given the past observations. For a given policy $\pi$, let $k_T^\pi$ denote the arm output at the end of $T$ budget. Let $\Pi$ denote the set of all policies that output an arm within $T$ budget on unimodal bandits instances.

**Objective:** The goal is to find a policy in $\Pi$ that exploits the unimodal structure of the mean rewards and minimizes the probability that the arm output at the end of $T$ budget is not the optimal arm. Specifically, our objective is given as follows:

$$\inf_{\pi \in \Pi} \sup_{\boldsymbol{\mu} \in \mathcal{S}} P_{\boldsymbol{p}(\boldsymbol{\mu})} \left( k_T^\pi \neq k^* \right), \tag{1}$$

where $\Pr(\cdot)$ is over the randomness of the reward and the policy. We refer to the BAI setup given in (1) as the *fixed budget BAI for unimodal bandits*.

### 3.3 Problem Dependent Complexity

For a given instance $\boldsymbol{p}(\boldsymbol{\mu}) \in \epsilon_U$ with arm means $\boldsymbol{\mu}$, we express the complexity as $\bar{H} := \bar{H}\big(\boldsymbol{p}(\boldsymbol{\mu})\big)$, and is given by

$$\bar{H} := \sum_{k \in \{k^*-1, k^*+1\}} \frac{1}{(\mu_{k^*} - \mu_k)^2}. \tag{2}$$

We call $\bar{H}$ complexity as the characterisation of the hardness of understanding the problem, as we will see later. Similar problem-dependent quantities are consider in Audibert et al. (2010); Carpentier & Locatelli (2016); Jamieson & Nowak (2014), that characterize the complexity of bandit problems, e.g.,

$$H_1 = \sum_{k \neq k^*} \frac{1}{(\mu_{k^*} - \mu_k)^2} \text{ and } H_2 = \sup_{k > 1} \frac{k}{(\mu_{k^*} - \mu_{(k)})^2},$$

where $\mu_{(k)}$ denotes the $k^{th}$ largest mean of the arms. Following Audibert et al. (2010), it is easy to show the following inequalities hold (see Appendix A.1 for proof)

$$\min(H_2, \bar{H}) \leq H_1 \leq 2 \log(K) H_2. \tag{3}$$

We next consider the lower bound for fixed budget BAI with the unimodal structure.

## 4 Lower Bound for Fixed Budget BAI of Unimodal Bandits

A lower bound on the error probability for BAI in the fixed budget setting without assuming any structure is established in Audibert et al. (2010); Garivier & Kaufmann (2016) and Carpentier & Locatelli (2016) using different techniques for constructing bandit problems, and all of them have provided minimax lower bound for the unstructured bandits. More specifically, Audibert et al. (2010) constructs $K!$ bandit problems by

permutation of the arms and proposes that for any bandit problem, there exists a permutation such that any algorithm will make an error with probability at least $\exp\left(-\frac{T}{H_2}\right)$. Garivier & Kaufmann (2016) proposed the 'flipping constructions' and showed that there exists a bandit problem such that any algorithm will make an error with probability at least $\exp\left(-\frac{T}{H_1}\right)$, where $H_2 < H_1$. Carpentier & Locatelli (2016) improved the flipping construction of Garivier & Kaufmann (2016) by providing further information to the algorithm. They proposed that there exists a bandit problem such that any algorithm will make an error with a probability of at least $\exp\left(-\frac{T}{\log_2(K)H_1}\right)$. The authors argue that in the fixed budget setting, unlike in the fixed confidence setting, there is an additional $\log(K)$ price to pay for adaptation to $H_1$ in the absence of knowledge over this quantity.

We adopt the lower bound proof of Carpentier & Locatelli (2016) for fixed budget BAI problems on unimodal instances. Carpentier & Locatelli (2016) provided a lower bound using a particular choice of Bernoulli rewards, whereas we do not assume any particular choice of bandit instances. Our only assumption is that an unimodal structure exists over the mean rewards, and we consider Gaussian distributions for our case. Below, we have given an overview of our construction.

Fix $\beta = 1$. Let $\boldsymbol{p}(\boldsymbol{\mu}) := \{p(\mu_k)\}_{k \in \mathcal{A}} \in \epsilon_U$ be a unimodal bandit instance such that $p(\mu_k) := N(\mu_k, 1)$, where $\mu_k \in [1/4, 1/2]$ for all $k \in \mathcal{A}$ and $\mu_{k^*} = 1/2$. Let $\boldsymbol{p}'(\boldsymbol{\mu}') := \{p'(\mu'_k)\}_{k \in \mathcal{A}}$ be another bandit instance where $p'(\mu'_k) := N(\mu'_k, 1)$ and $\mu'_k = 2\mu_{k^*} - \mu_k$ for all $k \in \mathcal{A}$. Using $\boldsymbol{p}(\boldsymbol{\mu})$ and $\boldsymbol{p}'(\boldsymbol{\mu}')$ we construct two more bandit instance $\mathbf{p}^{k^*-1}(\boldsymbol{\mu}^{k^*-1})$ and $\mathbf{p}^{k^*+1}(\boldsymbol{\mu}^{k^*+1})$ with means $\boldsymbol{\mu}^{k^*-1}$ and $\boldsymbol{\mu}^{k^*+1}$ as follows.

$$\mu_i^{k^*-1} = \mu_i \ \forall i \neq k^* - 1 \text{ and } \mu_i^{k^*-1} = \mu'_i \text{ for } i = k^* - 1$$
$$\mu_i^{k^*+1} = \mu_i \ \forall i \neq k^* + 1 \text{ and } \mu_i^{k^*-1} = \mu'_i \text{ for } i = k^* + 1$$

It is easy to note that both bandit instances $\mathbf{p}^{k^*-1}(\boldsymbol{\mu}^{k^*-1})$ and $\mathbf{p}^{k^*+1}(\boldsymbol{\mu}^{k^*+1})$ are unimodal with the optimal arms being $k^* - 1$ and $k^* + 1$, respectively. Recall the definition of $\bar{H}(\boldsymbol{p}(\boldsymbol{\mu}))$ given in (2). We define $\bar{h}$ as

$$\bar{h} := \sum_{i \in \{k^*-1, k^*+1\}} \frac{1}{(\mu_{k^*} - \mu_i)^2 \bar{H}\left(\mathbf{p}^i(\boldsymbol{\mu}^i)\right)},$$

where $\bar{H}\left(\mathbf{p}^i(\boldsymbol{\mu}^i)\right)$ for bandit instance $\mathbf{p}^i(\boldsymbol{\mu}^i)$ is defined as

$$\bar{H}\left(\mathbf{p}^i(\boldsymbol{\mu}^i)\right) = \sum_{k \in \{i-1, i+1\}} \frac{1}{(\Delta_k^i)^2}, \text{ where } \Delta_k^i = \begin{cases} 2\mu_{k^*} - \mu_i - \mu_k, & \text{if } k \neq i, \\ \mu_{k^*} - \mu_i, & \text{if } k = i. \end{cases}$$

Note that the authors in Carpentier & Locatelli (2016) have defined the quantity $h^* = \sum_{i \in \mathcal{A}, i \neq k^*} \frac{1}{(\mu_{k^*} - \mu_i)^2 \bar{H}(\mathbf{p}(\boldsymbol{\mu}^i))}$. However, for the unimodality structure of the mean rewards, we can define the quantity $\bar{h}$ on the neighbourhood of the optimal arm $k^*$. We now provide the lower bound of the fixed budget BAI problem for unimodal bandits, as stated in the following theorem.

**Theorem 1.** *For any unimodal bandit strategy that returns arm $k_T$ after $T$ budget, where $T \geq \max_{i \in \{k^*-1, k^*+1\}} \left(\bar{H}(\mathbf{p}(\boldsymbol{\mu})), \bar{H}(\mathbf{p}^i(\boldsymbol{\mu}^i))\bar{h}\right)^{\frac{4\log(6TK)}{12}}$, it holds that*

$$\max_{i \in \{k^*-1, k^*+1\}} \left[P_{\boldsymbol{p}^i(\boldsymbol{\mu}^i)}(k_T \neq i) \exp\left(15\frac{T}{\bar{H}(\mathbf{p}^i(\boldsymbol{\mu}^i))}\right)\right] \geq \frac{1}{6}. \tag{4}$$

*Proof.* The proof is given in Appendix A.2. $\square$

From the above theorem, we conclude that the lower bound for the fixed budget unimodal bandit is $O\left(\frac{1}{6}\exp\left\{-\frac{15T}{\bar{H}(\boldsymbol{p}(\boldsymbol{\mu}))}\right\}\right)$. Note that the exponent does not depend on $K$, but only on the sub-optimal gaps of

the neighbours of the optimal arm. Our focus on improving the scaling in $K$ is motivated by the study of unimodal bandits in the regret setting, where the unimodal property helps improve the scaling with respect to $K$. Specifically, a similar observation is also made for unimodal bandits in the cumulative regret setting, see Combes & Proutiere (2014).

## 5 FB-BAUB Algorithm

We propose an algorithm for unimodal bandit instances in the fixed budget BAI setting. The algorithm is based on the *Line Search Elimination (LSE)* method developed in Yu & Mannor (2011)and we refer to it as *Fixed Budget Best Arm in Unimodal Bandits (FB-BAUB)*. It is a parameter-free algorithm that only needs to know $K$ and $T$, and the arms are eliminated based on their empirical means.

FB-BAUB splits the total budget $T$ into $L + 1$ phases. Let $N_l$ for phase $l = 1, 2, \ldots, L + 1$, denotes the number of samples in phase $l$, where $\sum_{l=1}^{L+1} N_l = T$. We have chosen $N_l$ such that after the first two phases, the number of samples increases by a factor of $3/2$ in each subsequent phase, which helps to distinguish between the empirical means of the remaining arms, which are likely to be closer. Thus, $N_l$ is given by

$$N_l = \begin{cases} \frac{2^{L-2}}{3^{L-1}}T & \text{for } l = 1, 2 \\ \frac{2^{L-(l-1)}}{3^{L-(l-2)}}T & \text{for } l = 3, 4, \ldots, L + 1 \end{cases} \tag{5}$$

and satisfy the budget constraints, i.e,

$$2 \times \frac{2^{L-2}T}{3^{L-1}} + \sum_{l=3}^{L+1} \frac{2^{L-(l-1)}T}{3^{L-(l-2)}} = T. \tag{6}$$

The arms are sampled and eliminated in each phase, so only one arm survives after the $L + 1$ phase.

The pseudo-code of FB-BAUB is given in ALGO 1. It works as follows: Let $\mathcal{B}_l$ denote the set of arms available in phase $l$ and $j_l := |\mathcal{B}_l|$ is the number of arms in the set $\mathcal{B}_l$. In each phase $l = 1, 2, \ldots L$, the algorithm selects four arms $S_l = \{k^M, k^A, k^B, k^N\} \in \mathcal{B}_l$, which include the first, last and the two middle arms uniformly spaced from them (lines 4-7). Each of the arms is sampled for $N_l/4$ number of times (line 8). At the end of the phase, their empirical means, denoted as $\hat{\mu}_k$ (line 9), are obtained as follows:

$$\hat{\mu}_k^l = \frac{1}{N_l/4} \sum_{s=1}^{N_l/4} X_{k,s}^l, \quad \forall k \in S_l, \tag{7}$$

where $X_{k,s}^l$ denotes the $s^{th}$ sample from the $k^{th}$ arm in phase $l$. Based on these empirical means, we eliminate at most $1/3^{rd}$ of the number of arms from the remaining set.[1] Specifically, if the arms $k^M$ or $k^A$ have the highest empirical means, we eliminate all the arms succeeding $k^B$ in the set $\mathcal{B}_l$ (line 12). Similarly, if the arms $k^B$ or $k^N$ have the highest empirical means, we eliminate all the arms preceding $k^A$ in the set $\mathcal{B}_l$ (line 14). Fig. 1 gives a pictorial representation of the elimination of arms in two possible cases. The remaining set of arms is then transferred to the next phase. In phase $L + 1$, we are left with three arms. Each is sampled $N_{L+1}/3$ times, and the one with the highest empirical mean is output as the optimal arm (lines 18-22).

**Remark 1.** *Arms between $k^M$ & $k^A$ or $k^B$ & $k^N$ are eliminated in each phase, and the arms between $k^A$ & $k^B$ always survive.*

After phase $l = 1, 2, \ldots, L$, $\lfloor \frac{2}{3}j_l \rfloor$ of the arms survive. For ease of exposition, we will drop the $\lfloor \rfloor$ function since this drop will influence only a few constants in the analysis. Thus, after the end of the $L$ phases, there will be three arms as

$$(2/3)^L K = 3 \implies L = \frac{\log_2 K/3}{\log_2 3/2}. \tag{8}$$

Therefore, FB-BAUB outputs the best arm as $\hat{k}_{L+1}$ (i.e., $k_T$) after exploring for $T$ rounds.

---

[1] If the number of arms in a phase is not a multiple of 4, then less than $1/3^{rd}$ will be eliminated in that phase.

---

**ALGO 1: Fixed Budget Best Arm in Unimodal Bandits (FB-BAUB)**

---

1: **Input:** $T$ and $K$.
2: **Initialise:** $\mathcal{B}_1 = \mathcal{A}$, $j_1 \leftarrow |\mathcal{B}_1|$. Calculate $L$ from (8).
3: **for** $l = 1$ to $L$ **do**
4:     $k^M \leftarrow$ First arm of $\mathcal{B}_l$;
5:     $k^N \leftarrow$ Last arm of $\mathcal{B}_l$;
6:     $k^A \leftarrow \lceil j_l/3 \rceil^{th}$ arm of $\mathcal{B}_l$;
7:     $k^B \leftarrow \lfloor 2j_l/3 \rfloor^{th}$ arm of $\mathcal{B}_l$;
8:     Sample each arm in $S_l = \{k^M, k^A, k^B, k^N\}$ for $\frac{N_l}{4}$ number of times from (5)
9:     Obtain $\hat{\mu}^l_{k^M}, \hat{\mu}^l_{k^A}, \hat{\mu}^l_{k^B}, \hat{\mu}^l_{k^N}$ by (7).
10:    $x_l^* = \arg\max_{k \in S_l} \hat{\mu}_k$.
11:    **if** $x_l^* == \{k^M, k^A\}$ **then**
12:        $\mathcal{B}_{l+1} \leftarrow \{k \in \mathcal{B}_l : k^M \leq k \leq k^B\}$ Shrink to left
13:    **else if** $x_l^* == \{k^B, k^N\}$ **then**
14:        $\mathcal{B}_{l+1} \leftarrow \{k \in \mathcal{B}_l : k^A \leq k \leq k^N\}$ Shrink to right
15:    **end**
16:    $j_{l+1} \leftarrow |\mathcal{B}_{l+1}|$;
17: **end for**
18: **for** $l = L+1$ **do**
19:    $\mathcal{B}_{L+1} = \{k^M, k^A, k^N\}$;
20:    Sample each arm in $\{k^M, k^A, k^N\}$ for $\frac{N_{L+1}}{3}$ no. of times and obtain $\hat{\mu}^l_{k^M}, \hat{\mu}^l_{k^A}, \hat{\mu}^l_{k^N}$.
21:    Obtain $\hat{k}_{L+1} = \arg\max_{k \in \mathcal{B}_{L+1}} \hat{\mu}_k$.
22: **end for**
23: **Output:** $k_T = \hat{k}_{L+1}$

---

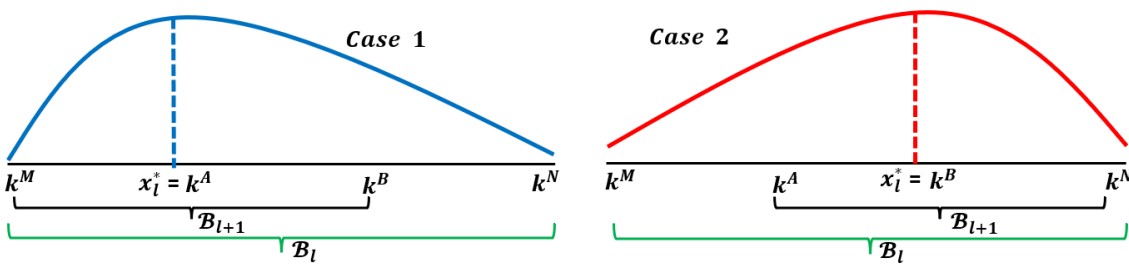

Figure 1: Different cases of elimination in phase $l$.

## 5.1 Comparison between LSE and FB-BAUB

In Yu & Mannor (2011), LSE is developed for a continuous set of arms. However, they have also applied LSE for a finite set of arms in the fixed confidence setting. FB-BAUB and LSE differ primarily in four aspects.

1. *Elimination:* LSE eliminates about $1 - 1/\phi$ fraction of arms, where $\phi$ is the golden ratio. In contrast, FB-BAUB eliminates $1/3^{rd}$ of the available arms in each phase.

2. *Input parameters:* LSE needs a sequence of parameters $(\epsilon_l, \delta_l)$ for every phase as input, whereas no such input parameters are required in FB-BAUB. Hence, it is a parameter-free algorithm, which is highly desirable.

3. *Number of samples*: In the discrete case, LSE considers the instance where mean rewards of neighboring arms are separated at least by an amount $D_L$, i.e., $\Delta > D_L$ (Yu & Mannor (2011)[Assum 3.4]), and uses this information in deciding the arm plays in each phase (Combes & Proutiere (2014)[Prop. 5.4]). Whereas FB-BAUB does not require any problem-specific information and works as long means of the neighboring arms are separated, i.e., $\Delta > 0$.

4. *Arms selection policy*: LSE adds one new arm based on the golden ratio in each phase. Whereas FB-BAUB adds two new arms in each phase by uniformly dividing the space.

# 6 Performance Guarantee of FB-BAUB

This section provides the following theorem that gives the upper bound for the error probability of FB-BAUB.

**Theorem 2.** *Let $\boldsymbol{p}(\boldsymbol{\mu}) \in \epsilon_U$ follow $\beta$ sub-Gaussian with arm means $\boldsymbol{\mu}$ and $\Delta = \min\limits_{2 \leq i \leq K} |\mu_i - \mu_{i-1}| > 0$ denote the minimum gap between the means of any two neighboring arms. For any $T > K$, the error probability of FB-BAUB is bounded as*

$$
P_{\boldsymbol{p}(\boldsymbol{\mu})}(\hat{k}_{L+1} \neq k^*) \leq 2 \exp\left\{-\frac{TK}{32}\left(\frac{\Delta}{\beta}\right)^2\right\} + 2 \exp\left\{-\frac{TK}{72}\left(\frac{\Delta}{\beta}\right)^2\right\}
$$
$$
+ 2 \exp\left\{-\frac{T}{24}\left(\frac{\Delta}{\beta}\right)^2\right\} + 2(L-2) \exp\left\{-\frac{T}{8}\left(\frac{\Delta}{\beta}\right)^2\right\}. \tag{9}
$$

*Proof.* The proof is given in Appendix A.3. □

The first two terms in the upper bound correspond to the probability of eliminating the optimal arm $k^*$ in phases 1 & 2. The $3^{rd}$ term bounds the probability of eliminating the optimal arm in phase $L + 1$, and the $4^{th}$ term corresponds to the sum of the probabilities of eliminating the optimal arm in phases $l = 3, \ldots, L$. Note that, as $K > 1$, $\exp\left\{-TK\left(\frac{\Delta}{\beta}\right)^2\right\} < \exp\left\{-T\left(\frac{\Delta}{\beta}\right)^2\right\}$. As a result, the $3^{rd}$ and $4^{th}$ terms dominate the upper bound (9). Therefore, the error probability is of order $\mathcal{O}\left(\log_2 K \exp\left\{-T\Delta^2\right\}\right)$, where the error exponent term $\exp\left\{-T\Delta^2\right\}$ does not depend on $K$.

**Comparison with Sequential Halving:** For unstructured bandits, the complexity parameter $H_2$, as given in Subsection 3.3, can be upper bounded by $H_2 < H_1 = \sum\limits_{k \neq k^*} \frac{1}{(\mu_{k^*} - \mu_k)^2} \leq \sum\limits_{k \neq k^*} \frac{1}{(k - k^*)^2 \Delta^2} \leq \frac{\pi^2}{3\Delta^2} \leq \frac{4}{\Delta^2}$. Applying this inequality, the upper bound of the error probability of *Sequential Halving* as proposed in Karnin et al. (2013)[Thm. 4.1] is revised as of order $O\left(\log_2 K \exp\left\{-\frac{T\Delta^2}{\log K}\right\}\right)$ and is optimal as it matches with the lower bound derived in Carpentier & Locatelli (2016) up to a multiplicative factor of $\log_2 K$. Note that the exponent term in the error bound of Sequential Halving has a $\log_2 K$ factor. For unimodal bandits, the exponent term in the error bound of FB-BAUB does not depend on $K$ and is smaller by a factor of $\log_2 K$. As expected, the error probability for unimodal bandits should be smaller, and our analysis captures this gain by shaving off the factor $\log_2 K$ in the error bound.

**Comparision with LSE:** The following points will discuss the novelty of the analysis of FB-BAUB, compared to the analysis of LSE, as follows:

1. **Technical Challenges of FB-BAUB:** The PAC-bound provided by the LSE algorithm in Yu & Mannor (2011) is based on the known $(\epsilon_l, \delta_l)$-PAC bound of the Sampling Algorithm (refer to Thm. 4.1) for every iteration. However, FB-BAUB does not run any sub-algorithm but has to carefully construct $N_l$ so that the budget constraint that balances the trade-off of elimination and exploration of new arms is attained. Note that $N_l$ are of different lengths over the phases, and the error bound of the FB-BAUB is obtained by carefully applying Hoeffding's inequality in each phase.

2. **Minimum Gap Separation:** For a finite set of arms, LSE assumes that the gap between the mean rewards of the neighboring arms is separated by at least $D_L > 0$ (Assumption 3.2 in Yu & Mannor (2011)). More specifically, LSE requires knowledge of $D_L$, and its sample complexity is expressed in terms of $(\epsilon_l, \delta_l)$ for each phase $l$. However, FB-BAUB only considers that $\Delta > 0$, i.e., the arm means to be distinct. We do not need any assumptions on the minimum separation of the mean rewards of the neighboring arms, i.e., FB-BAUB need not know $D_L$. Therefore, FB-BAUB works when the arm means are distinct but arbitrarily close to each other, whereas LSE analysis assumes that this separation is at least $D_L$.

3. **Analysis of fixed-budget setting vs. fixed-confidence setting:** Our analysis for the finite set of arms differs from that of Yu & Mannor (2011), as we do not require the knowledge of $D_L$. Since $D_L$ is unknown, this is not the same as 'computing the sample size with a required estimation error'. We have to carefully control the elimination and decide the duration of the samples to meet the budget constraints. However, as rightly noted, in the case of Yu & Mannor (2011), 'computing the sample size with a required estimation error' applies as they use the knowledge of $D_L$ for the finite set of arms.

4. **Novelty in choice of $N_l$:** The nature of selecting an arm for sampling based on the golden ratio in Yu & Mannor (2011) makes the separation between the arms selected for sampling non-uniform, which in turn makes the number of arms eliminated in each phase a random quantity. Hence, LSE is not a good strategy for fixed-budget settings where the number of samples is constrained, and we need to have a good accounting of the remaining arms for analytical traceability. On the other hand, in FB-BAUB, 2/3 of the arms remain in each phase, and we increase the number of samples collected in subsequent phases by a factor of 3/2. This helps us in two ways. (a) meet the budget constraint exactly across the phases, and (b) the number of samples in phase independent of the problem instance (like knowing $D_L$).

We note that the error exponent of FB-BAUB differs from the optimal error exponent with respect to the complexity terms as $\bar{H}(\mathbf{p}(\boldsymbol{\mu})) \leq 2/\Delta^2$ and hence is not optimal with respect to the specific problem instance. It is an interesting open problem to develop an optimal algorithm in the fixed budget BAI setting with unimodality. However, our main focus is to improve the scaling of the error probability bound w.r.t. $K$. We have provided a near-optimal solution and have shown that the error exponent for both the lower and upper bounds does not involve $K$, quantifying the gain achieved by exploiting the unimodal structure. We are motivated by the study of unimodal bandits in the regret setting (Combes & Proutiere (2014)), where the unimodal property helps improve the scaling of the regret bound with respect to $K$.

# 7 Simulation Results

In this section, we corroborate the theoretical guarantee of FB-BAUB by applying it to problem instances of varying difficulty. As no algorithm exists for the BAI in unimodal bandits in the fixed budget setting, we consider the following benchmark algorithms. The source code is available at https://github.com/debamita-ghosh/FBBAUB.

**Sequential Halving (Seq. Halv.) Karnin et al. (2013):** This fixed budget BAI algorithm is for unstructured bandits but is shown to be optimal by Carpentier & Locatelli (2016). A comparison with this algorithm gives gains achieved by exploiting structure.

**Successive Rejects (SR) Audibert et al. (2010):** This fixed budget BAI algorithm for unstructured bandits is parameter-free and optimal up to a logarithmic term. SR outperforms UCB-E in Audibert et al. (2010); Shahrampour et al. (2017). Hence, we do not consider UCB-E for comparison. A comparison with this algorithm gives gains achieved by exploiting structure.

**Linear Search Elimination (LSE) Yu & Mannor (2011):** We consider the discrete variant of LSE proposed for a finite set of arms in the fixed confidence case and adopt it to the fixed budget setting. A comparison of FB-BAUB with LSE is pertinent as it is a well-known algorithm for unimodal bandits.

## 7.1 Comparison with pure exploration algorithms for unimodal structure

We consider two experimental setups to compare the performance of FB-BAUB with other benchmark algorithms. We consider $K$ Gaussian arms with known variances $\sigma_i^2 = \sigma^2 = 5$, assuming that the mean of the best arm is $\mu_{k^*} = -252$. Our simulations are averaged over 1000 runs and are shown with confidence intervals.

**Experiment 1:** $\mu_1 = -312$, $\mu_{k^*} = -252$, $\mu_K = -311$, $\mu_{2:k^*-1} = \mu_1 + \frac{2(k-1)(\mu_{k^*}-\mu_1)}{k^*-1}$, and $\mu_{k^*+1:K} = \mu_{k^*} - \frac{2(k-k^*)(\mu_{k^*}-\mu_K)}{K-k^*-1}$.

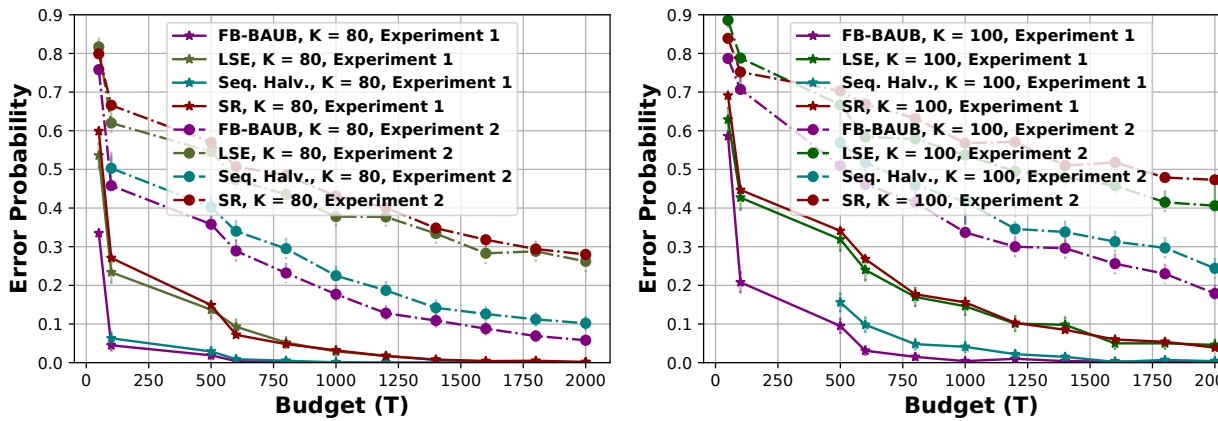

Figure 2: Error Probability vs $T$, $K = 80$.   Figure 3: Error Probability vs $T$, $K = 100$.

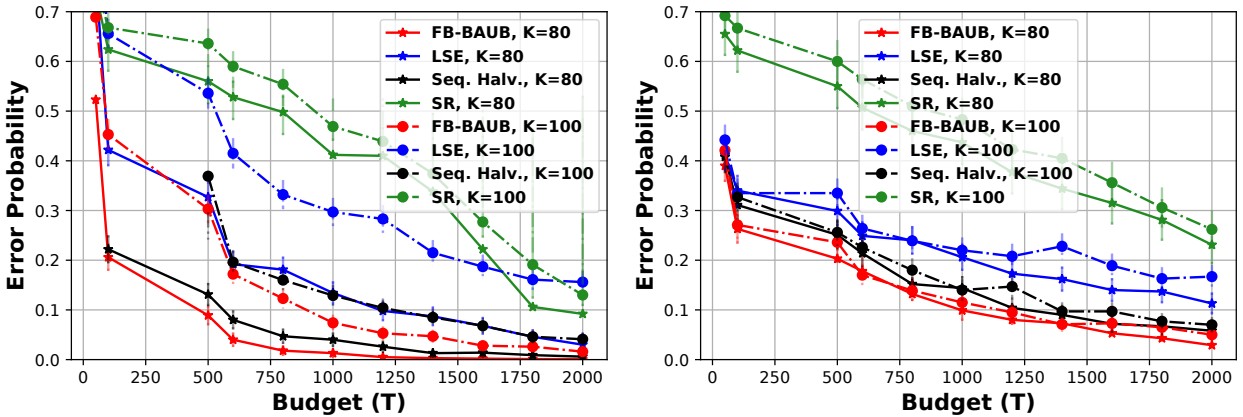

Figure 4: Exp. 3: Error Prob. vs $T$.   Figure 5: Exp. 4: Error Prob. vs $T$.

**Experiment 2:** $\mu_1 = -312$, $\mu_{k^*} = -252$, $\mu_K = -311$, $\mu_{2:k^*-1} = \mu_1 + \frac{(k-1)(\mu_{k^*}-\mu_1)}{(k^*-1)}$, and $\mu_{k^*+1:K} = \mu_{k^*} - \frac{(k-k^*)(\mu_{k^*}-\mu_K)}{(K-k^*-1)}$.

Fig. 2 and Fig. 3 illustrate the performance of each algorithm for Exp. 1 and Exp. 2. We examine each setup for two values of $K = \{80, 100\}$. Exp. 1 and 2 are strictly unimodal in the sense that the optimal arm is lying in the interior. In Exp. 1, the means of the successive arms are well-separated compared to that in Exp. 2, i.e., $\Delta$ is lower for Exp. 2. As we decrease the gap between the means of the neighboring arms, the means of the neighboring arms of the optimal arm come close to each other. Hence, identifying the best arm becomes complicated, which increases the error probability. This makes Exp. 2 more challenging to identify the optimal arm compared to Exp. 1, and thereby the error probability is higher for Exp. 2 than Exp. 1. Moreover, the error probability increases as we increase the arm size. We compared the empirical number of pulls of each arm for Exp. 1 and 2 for the state-of-the-art algorithms in Appendix A.4, see Fig. 7 and Fig. 8.

SR has the worst error performance for each setup. For a small number of arms, both Seq. Halv. and FB-BAUB have comparable performance, but as the number of arms increases, FB-BAUB has a lesser error probability compared to Seq. Halv. as evident from the case of $K = 100$. Moreover, FB-BAUB can identify the best arm with a probability of more than 95% with a lesser budget than other state-of-the-art algorithms. More specifically, in Exp. 1 *FB-BAUB* can identify the best arm with a probability of more than 95% within 100 budget for 80 arms, while the other state-of-the-art algorithms need at least a 500 budget for executions. Hence, FB-BAUB outperforms the state-of-the-art algorithms.

We note that the minimum budget requirement (as a function of $K$) for LSE is much smaller than both FB-BAUB and Seq. Halv. for its feasible execution. However, as LSE samples for a fixed number of times for each of the arms in every phase, the number of samples it runs for arms neighboring $k^*$ is much lesser, resulting in a higher error probability. Seq. Halv. needs at least $K \log_2(K)$ number of horizons to complete one phase and has samples for all arms in every phase. Note that for each setup, Seq. Halv. requires at least 100 and 300 rounds for $K = 80$ and $K = 100$, respectively, to complete their execution, and hence, their graph starts after those many rounds. Thereby, it has fewer rounds remaining to explore the best arm when the algorithm is executed in the neighbourhood of $k^*$ compared to FB-BAUB. Thus, the minimum budget requirement for FB-BAUB as a function of $K$ is much less than that of Seq. Halv. In addition, FB-BAUB has better error probability performance. This demonstrates the advantage of exploiting the unimodality of the reward function.

### 7.2 Comparison with pure exploration algorithms for monotone structure

We have considered two more experimental set-ups to compare FB-BAUB with the state-of-the-art algorithms. We consider $K$ Gaussian arms with known variances $\sigma_i^2 = \sigma^2 = 0.5$, assuming that the mean of the best arm is $\mu_1 = 0.7$. Our simulations are averaged over 1000 runs and are shown with confidence intervals.

**Experiment 3:** $\mu_1 = 0.7$ and $\mu_{2:K} = \mu_1 - \frac{0.6(i-1)}{K-1}$.

**Experiment 4:** $\mu_1 = 0.7$ and $\mu_{2:K} = \mu_1 - 0.01 \left(1 + \frac{4}{K}\right)^{i-2}$.

Fig. 4 and Fig. 5 illustrate the performance of each algorithm for Exp. 3 and Exp. 4, respectively, for two values of $K = \{80, 100\}$. Exp. 3 and Exp. 4 follow the monotone structure with the first arm as the optimal arm, similar to that considered in Shahrampour et al. (2017); Audibert et al. (2010). The error probability increases as we increase the arm size. In Exp. 3, the sub-optimal gap decreases in arithmetic progression, whereas in Exp. 4, the sub-optimal gap decreases in geometric progression. This makes Exp. 4 more challenging to find the best arm within a fixed budget, and thereby, the error probability of FB-BAUB is lower in Exp. 3 than in Exp. 4. We compared the empirical number of pulls of each arm for Exp. 3 and 4 for the state-of-the-art algorithms in Appendix A.4, see Fig. 9 and Fig. 10.

Furthermore, in Exp. 3, FB-BAUB can identify the optimal arm with a probability of more than 80% within 100 budget for 80 arms, and in Exp. 4, it identifies the optimal arm with a probability of more than 75% within 100 budget. Whereas, the other state-of-the-art algorithms need at least a 600 budget for execution. Hence, FB-BAUB outperforms the other state-of-the-art algorithms for Exp. 3 and 4.

## 8   Conclusion

We studied the fixed budget BAI problem with an unimodal structure on a finite set of $K$ arm means. We developed an algorithm named FB-BAUB to address the problem and derived an upper bound on its error probabilities. The algorithm works in phases and identifies the best arm with high probability. We established that the exponent in the error bound is independent of $K$ in contrast to the unstructured bandits. We demonstrated that for any optimal algorithm, the error exponent should be independent of $K$ by establishing a lower bound. Simulations validated the efficiency of FB-BAUB compared to state-of-the-art algorithms. FB-BAUB is parameter-free and easy to implement.

Many interesting research directions could be further investigated. The exponent in the upper bound on the error probability of FB-BAUB is optimal in $T$ and $K$, but not in the problem-dependent complexity terms, which are characterized in terms of the smallest gap between any neighbouring arms. However, the lower bound is only dependent on the neighbours of the optimal arm. It is interesting to develop algorithms that are also optimal with respect to the complexity terms.

## Acknowledgement

Manjesh K. Hanawal thanks funding support from SERB, Govt. of India, through the Core Research Grant (CRG/2022/008807) and MATRICS grant (MTR/2021/000645), and funding support from DST, Govt. of India, through the DST-INRIA targeted programme and the DST-INRIA joint research targeted programme.

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
