# A    Appendix

## A.1    Proof of Inequalities [Eq. (3)]

*Proof.* Note that in Audibert et al. (2010), the following inequality holds:

$$H_2 \leq H_1 \leq \log(2K)H_2 \leq 2\log(K)H_2 \tag{10}$$

By the definition of $\bar{H}$, we have

$$\bar{H} = \sum_{k \in \{k^*-1, k^*+1\}} \frac{1}{(\mu_{k^*} - \mu_k)^2} \leq \sum_{k \neq k^*} \frac{1}{(\mu_{k^*} - \mu_k)^2} = H_1 \quad \text{(By the definition of } H_1) \tag{11}$$

Combining (10) and (11) we obtain the given inequality in (3). □

## A.2    Proof of Theorem 1

First, we state the following Theorem and Corollary to prove Theorem 1.

**Theorem 3.** *For any bandit strategy that returns the arm $k_T$ after $T$ budget, it holds that*

$$\max_{i \in \{k^*-1, k^*+1\}} P_{\boldsymbol{p}^i(\boldsymbol{\mu}^i)}(k_T \neq i) \geq \frac{1}{6} \exp\left(-12\frac{T}{\bar{H}(k^*)} - 2\sqrt{12\frac{T}{\bar{H}(k^*)}\log(6TK)}\right), \tag{12}$$

*and also*

$$\max_{i \in \{k^*-1, k^*+1\}} \left[P_{\boldsymbol{p}^i(\boldsymbol{\mu}^i)}(k_T \neq i) \exp\left(12\frac{T}{\bar{h}\bar{H}(i)} + 2\sqrt{12\frac{T}{\bar{h}\bar{H}(i)}\log(6TK)}\right)\right] \geq \frac{1}{6}. \tag{13}$$

*Proof.* The proof of this theorem follows the lines similar to Carpentier & Locatelli (2016)[Thm. 2] after applying the change of measure on the restricted set of arms.

Fix $\beta = 1$. Let $\mathbf{p}(\boldsymbol{\mu}) := \{p(\mu_k)\}_{k \in \mathcal{A}} \in \epsilon_U$ be a unimodal bandit instance such that $p(\mu_k) := N(\mu_k, 1)$, where $\mu_k \in [1/4, 1/2]$ for all $k \in \mathcal{A}$ and $\mu_{k^*} = 1/2$. We also consider $\mu_1 < \mu_2 < \cdots < \mu_{k^*-1} < \mu_{k^*} > \mu_{k^*+1} > \cdots > \mu_K$. We would like to find the lower bound of the probability that the learner fails to recommend the optimal arm when presented with instance $\boldsymbol{\mu}$, i.e., $P(k_T \neq k^*)$.

We define $d_k := \mu_{k^*} - \mu_k = \frac{1}{2} - \mu_k$, for any $1 \leq k \leq K$. Set $\Delta_k^i = d_i + d_k$, if $k \neq i$ and $\Delta_i^i = d_i$, for any $i \in \{k^*-1, k^*+1\}$ and any $k \in \{1, \ldots, K\}$. Note that $\{\Delta_k^i\}_k$ denotes the arm gaps of the bandit problem $i$.

We adapt the proof of Carpentier & Locatelli (2016)[Thm. 2] to include the unimodal structure to derive a lower bound by applying the change of measure rule to on the restricted set of arms.

- **Step 1: The bandit problems that satisfy the unimodal structure**

  We have considered $K$ pairs of Gaussian arms where $\mathbf{p}(\boldsymbol{\mu}) := \{p(\mu_k)\}_{k \in \mathcal{A}} \in \epsilon_U$ be a unimodal bandit instance such that $p(\mu_k) := N(\mu_k, 1)$, where $\mu_k \in [1/4, 1/2]$ for all $k \in \mathcal{A}$ and $\mu_{k^*} = 1/2$, and $\mathbf{p}'(\boldsymbol{\mu}') := \{p'(\mu'_k)\}_{k \in \mathcal{A}}$ be another bandit instance where $p'(\mu'_k) := N(\mu'_k, 1)$ and $\mu'_k = 2\mu_{k^*} - \mu_k$ for all $k \in \mathcal{A}$.

  According to Carpentier & Locatelli (2016)[Thm. 2], we define $K$ Gaussian bandit problem using "flipping constructions" where for the bandit problem $\mathbf{p}^i(\boldsymbol{\mu}^i) := p^i(\mu_1^i) \otimes p^i(\mu_2^i) \otimes \cdots \otimes p^i(\mu_K^i)$ with means $\boldsymbol{\mu}^i$, arm $i$ is the optimal arm with distribution

$$p^i(\mu_k^i) = \begin{cases} p(\mu_k), & \text{if } i \neq k \\ p'(\mu'_k), & \text{if } i = k. \end{cases}$$

  Hence, the bandit problem $\mathbf{p}^i(\boldsymbol{\mu}^i)$ does not follow unimodal structure if $i \notin \{k^* - 1, k^*, k^* + 1\}$. Therefore, by flipping the distributions for all other arms will result in a non-unimodal bandit problem except for the bandit problems $\mathbf{p}^i(\boldsymbol{\mu}^i)$ where $i \in \{k^* - 1, k^*, k^* + 1\}$. For simplicity, we will denote bandit problem $\mathbf{p}^i(\boldsymbol{\mu}^i)$ as $i$. Note that the bandit problem $\mathbf{p}^{k^*}(\boldsymbol{\mu}^{k^*}) = \mathbf{p}(\boldsymbol{\mu})$.

  Hence we will focus on the three bandit problems $\mathbf{p}^i(\boldsymbol{\mu}^i)$ for $i \in \{k^* - 1, k^*, k^* + 1\}$ that follows unimodal structure. For $i \in [K]$, we use the notation $P_{\mathbf{p}^i(\boldsymbol{\mu}^i)}(.)$ and $E_{\mathbf{p}^i(\boldsymbol{\mu}^i)}(.)$ to denote the probability and expectation, respectively, with respect to the randomness of sampling for bandit problem $i$.

- **Step 2: Definition of high probability event and concentration of empirical KL divergences**

  For two distributions $p$ and $p'$ defined on $\mathbb{R}$ and $p$ is absolutely continuous with respect to $p'$, the Kullback Leibler (KL) divergence between distribution $p$ and $p'$, can be written as

$$\mathrm{KL}(p, p') = \int_{\mathbb{R}} \log\left(\frac{dp(x)}{dp'(x)}\right) dp(x),$$

  Following the lines of proof of Atsidakou et al. (2022)[Thm. 9], for $k \in \{1, 2, \ldots, K\}$, the KL divergence between two Gaussian distributions $p(\mu_k)$ and $p'(\mu'_k)$ is given by

$$\mathrm{KL}_k := \mathrm{KL}\left(p(\mu_k), p'(\mu'_k)\right) = \frac{(\mu_k - \mu'_k)^2}{2} = 2d_k^2. \tag{14}$$

  Let us consider $1 \leq t \leq T$. We define the quantity as

$$\widehat{\mathrm{KL}}_{k,t} = \frac{1}{t} \sum_{s=1}^{t} \log\left(\frac{dp(\mu_k)}{dp'(\mu'_k)}(X_{k,s})\right) = \frac{1}{t} \sum_{s=1}^{t} 2(X_{k,s} - \mu_{k^*})d_k$$

  where $X_{k,s}$ are independently and identically (i.i.d.) distributed as $p^i(\mu_k^i)$ for $s \leq t$ and bandit problem $i$.

  Note that

$$\mathbb{E}_{\boldsymbol{p}^i(\boldsymbol{\mu}^i)}\left[\widehat{KL}_{k,t}\right] = \begin{cases} 2(\mu_k - \mu_{k^*})d_k = -KL_k, k \neq i \\ 2(\mu_k + 2d_k - \mu_{k^*})d_k = KL_k, k = i \end{cases}$$

  This implies that $\left|\widehat{KL}_{k,t}\right|$ is an unbiased estimator of $KL_k$.

  Let us define an event as follows:

$$\zeta = \left\{ \forall 1 \leq k \leq K, \forall 1 \leq t \leq T, \left|\widehat{\mathrm{KL}}_{k,t}\right| - \mathrm{KL}_k \leq 2d_k\sqrt{\frac{2\log(6TK)}{t}} \right\}. \tag{15}$$

According to Carpentier & Locatelli (2016)[Lemma 1], the concentration bound for $\left|\widehat{KL}_{k,t}\right|$ that holds for the bandit problem $i$ where $i \in \{k^* - 1, k^*, k^* + 1\}$ is given by,

$$P_{\boldsymbol{p}^i(\boldsymbol{\mu}^i)}(\zeta) \geq \frac{5}{6}, \quad \text{for } i \in \{k^* - 1, k^*, k^* + 1\}. \tag{16}$$

- **Step 3: A change of measure**

Let *Alg* denote the active strategy of the learner that returns some arm $k_T$ at the end of the budget $T$. Let $\{T_k\}_{1 \leq k \leq K}$ denote the numbers of samples collected by *Alg* on each arm of the bandits, and they are stochastic in nature. Note that according to the definition of the fixed budget setting, we have $\sum_{1 \leq k \leq K} T_k = T$.

Let us write for any $0 \leq k \leq K$,

$$t_k = \mathbb{E}_{\boldsymbol{p}^{k^*}(\boldsymbol{\mu}^{k^*})}\left[T_k\right] \quad \text{and} \quad \sum_{1 \leq k \leq K} t_k = T.$$

We recall the change of measure identity, refer Audibert et al. (2010), which states that for any measurable event $\xi$ and for any $i \in \{k^* - 1 k^* + 1\}$, we have

$$P_{\boldsymbol{p}^i(\boldsymbol{\mu}^i)}(\xi) = \mathbb{E}_{\boldsymbol{p}^{k^*}(\boldsymbol{\mu}^{k^*})}\left[1\{\xi\} \exp\left(-T_i \widehat{KL}_{i,T_i}\right)\right] \tag{17}$$

as the product distributions $\mathbf{p}^i(\boldsymbol{\mu}^i)$ and $\mathbf{p}^{k^*}(\boldsymbol{\mu}^{k^*})$ only differ in arm $i$ and as the active strategy only explored the samples $\{X_{k,s}\}_{k \leq K, s \leq T_k}$

We now consider the event $\xi_i$ as the event where the algorithm outputs arm $k^*$ at the end of $T$ budget, where $\zeta$ holds, and where the number of times arm $i$ was pulled is smaller than $6t_i$, i.e.,

$$\xi_i = \left\{k_T = k^*\right\} \cap \left\{\zeta\right\} \cap \left\{T_i \leq 6t_i\right\}, \tag{18}$$

for $i \in \{k^* - 1, k^* + 1\}$. Applying the event $\xi_i$ as given by (18) in (17) we obtain,

$$P_{\boldsymbol{p}^i(\boldsymbol{\mu}^i)}(\xi) = \mathbb{E}_{\boldsymbol{p}^{k^*}(\boldsymbol{\mu}^{k^*})}\left[1\{\xi_i\} \exp\left(-T_i \widehat{KL}_{i,T_i}\right)\right]$$

Following the same lines of proof as given in Carpentier & Locatelli (2016)[Step 2, Thm. 2] and in Atsidakou et al. (2022)[Lemma 17] and applying (14), for $i \in \{k^* - 1, k^* + 1\}$, we get

$$P_{\boldsymbol{p}^i(\boldsymbol{\mu}^i)}(\xi_i) \geq P_{\boldsymbol{p}^{k^*}(\boldsymbol{\mu}^{k^*})}(\xi_i) \exp\left(-12t_i d_i^2 - 2\sqrt{12t_i d_i^2 \log(6TK)}\right). \tag{19}$$

- **Step 4: Lower bound on $\mathbb{P}_{\boldsymbol{p}^{k^*}(\boldsymbol{\mu}^{k^*})}(\xi_i)$ for any reasonable algorithm**

Let us assume that the probability that *Alg* makes a mistake on problem $k^*$ is less than $1/2$, i.e.,

$$\mathbb{E}_{\boldsymbol{p}^{k^*}(\boldsymbol{\mu}^{k^*})}\left[k_T \neq k^*\right] \leq \frac{1}{2} \tag{20}$$

Note that if *Alg* does not satisfy that, it performs badly on bandit problem $k^*$, and its probability of success is not larger than $\frac{1}{2}$ uniformly on the three bandit problems we defined for $\{k^* - 1, k^*, k^* + 1\}$.

For any $1 \leq k \leq K, k \neq k^*$ it holds by Markov's inequality that

$$P_{\boldsymbol{p}^{k^*}(\boldsymbol{\mu}^{k^*})}(T_k \geq 6t_k) \leq \frac{\mathbb{E}_{\boldsymbol{p}^{k^*}(\boldsymbol{\mu}^{k^*})}[T_k]}{6t_k} = \frac{1}{6}, \tag{21}$$

since $\mathbb{E}_{\boldsymbol{p}^{k^*}(\boldsymbol{\mu}^{k^*})}[T_k] = t_k$ for Algorithm *Alg*.

Therefore, by combining (20), (21) and (16), it holds by an union bound that for any $i \in \{k^*-1, k^*+1\}$

$$P_{\boldsymbol{p}^{k^*}(\boldsymbol{\mu}^{k^*})}(\xi_i) \geq 1 - \left(\frac{1}{6} + \frac{1}{2} + \frac{1}{6}\right) = \frac{1}{6}. \tag{22}$$

We will now combine (22) and the fact that $P_{\boldsymbol{p}^i(\boldsymbol{\mu}^i)}(k_T \neq i) \geq P_{\boldsymbol{p}^i(\boldsymbol{\mu}^i)}(\xi_i)$ for $i \in \{k^*-1, k^*, k^*+1\}$ and by applying in (17), we obtain

$$P_{\boldsymbol{p}^i(\boldsymbol{\mu}^i)}(k_T \neq i) \geq \frac{1}{6} \exp\left(-12 t_i d_i^2 - 2\sqrt{12 t_i d_i^2 \log(6TK)}\right) \tag{23}$$

- **Step 5: Conclusions**

  We defined $\bar{H}(k^*) := \sum_{k \in \{k^*-1, k^*+1\}} \frac{1}{(\Delta_k^{k^*})^2}$. We also know that $\sum_{1 \leq k \leq K} t_k = T$. By combining these two facts, we can say that there exists $i \in \{k^*-1, k^*+1\}$ such that

  $$t_i \leq \frac{T}{\bar{H}(k^*) d_i^2}$$

  as the contraposition yields an immediate contradiction. For this $i$, it holds by (23) that

  $$\max_{i \in \{k^*-1, k^*+1\}} P_{\boldsymbol{p}^i(\boldsymbol{\mu}^i)}(k_T \neq i) \geq \frac{1}{6} \exp\left(-12 \frac{T}{\bar{H}(k^*)} - 2\sqrt{12 \frac{T}{\bar{H}(k^*)} \log(6TK)}\right). \tag{24}$$

  This concludes the proof of the first part of the theorem.

  We have, $\mu_k = \frac{1}{2} - d_k$ such that $\mu_k \in [1/4, 1/2], \forall k \in \mathcal{A}$ and $\mu_{k^*} = \frac{1}{2}$. Note that $\boldsymbol{\mu}$ exhibits a unimodal structure. Since $\bar{h} = \sum_{i \in \{k^*-1, k^*+1\}} \frac{1}{d_i^2 \bar{H}(i)}$ and since $\sum_{1 \leq k \leq K} t_k = T$, then there exists $i \in \{k^*-1, k^*+1\}$ such that

  $$t_i \leq \frac{T}{\bar{h} \bar{H}(i) d_i^2}$$

  Therefore, for these $i \in \{k^*-1, k^*+1\}$ and by (23) we get

  $$\max_{i \in \{k^*-1, k^*+1\}} \left[P_{\boldsymbol{p}^i(\boldsymbol{\mu}^i)}(k_T \neq i) \exp\left(12 \frac{T}{\bar{h} \bar{H}(i)} + 2\sqrt{12 \frac{T}{\bar{h} \bar{H}(i)} \log(6TK)}\right)\right] \geq \frac{1}{6}. \tag{25}$$

  This concludes the proof of the second part of the theorem. $\qquad \square$

**Corollary 1.** *Assume that* $T \geq \max_{i \in \{k^*-1, k^*+1\}} \left(\bar{H}(k^*), \bar{H}(i)\bar{h}\right) \frac{4 \log(6TK)}{12}$. *For any bandit strategy that returns the arm* $\hat{k}_T$ *at time* $T$, *it holds that*

$$\max_{i \in \{k^*-1, k^*+1\}} P_{\boldsymbol{p}^i(\boldsymbol{\mu}^i)}(k_T \neq i) \geq \frac{1}{6} \exp\left(-24 \frac{T}{\bar{H}(k^*)}\right),$$

*and also*

$$\max_{i \in \{k^*-1, k^*+1\}} \left[P_{\boldsymbol{p}^i(\boldsymbol{\mu}^i)}(k_T \neq i) \exp\left(24 \frac{T}{\bar{H}(i)\bar{h}}\right)\right] \geq \frac{1}{6}.$$

*Proof.* We assumed that

$$T \geq \max_{i \in \{k^*-1, k^*+1\}} \left(\bar{H}(k^*), \bar{H}(i)\bar{h}\right) \frac{4 \log(6TK)}{12}. \tag{26}$$

- **Case 1:** Let us consider

$$\max\left(\bar{H}(k^*), \bar{H}(i)\bar{h}\right) = \bar{H}(k^*) \tag{27}$$

Applying (27) in (26) we obtain,

$$\frac{12T}{\bar{H}(k^*)} \geq 2\sqrt{12\frac{T}{\bar{H}(k^*)}\log(6TK)} \tag{28}$$

Applying (28) in (24) we get

$$\max_{i\in\{k^*-1,k^*+1\}} P_{\boldsymbol{p}^i(\boldsymbol{\mu}^i)}(k_T \neq i) \geq \frac{1}{6}\exp\left(-24\frac{T}{\bar{H}(k^*)}\right) \tag{29}$$

This concludes the proof of the first part of the corollary.

- **Case 2:** Let us consider for each $i \in \{k^*-1, k^*+1\}$

$$\max\left(\bar{H}(k^*), \bar{H}(i)\bar{h}\right) = \bar{H}(i)\bar{h} \tag{30}$$

Applying (30) in (26) we obtain,

$$\frac{12T}{\bar{h}\bar{H}(i)} \geq 2\sqrt{12\frac{T}{\bar{h}\bar{H}(i)}\log(6TK)} \tag{31}$$

Applying (31) in (25) we get

$$\max_{i\in\{k^*-1,k^*+1\}}\left[P_{\boldsymbol{p}^i(\boldsymbol{\mu}^i)}(k_T \neq i)\exp\left(24\frac{T}{\bar{H}(i)\bar{h}}\right)\right] \geq \frac{1}{6}. \tag{32}$$

This concludes the proof of the second part of the corollary. $\qquad\square$

We will now prove Theorem 1 using this corollary.

*Proof.* We have $\bar{h}$ defined as

$$\bar{h} = \sum_{i\in\{k^*-1,k^*+1\}} \frac{1}{d_i^2\bar{H}(i)}$$

Let us denote

$$(I) := \frac{1}{d_{k^*-1}^2\bar{H}(k^*-1)} \tag{33}$$

$$(II) := \frac{1}{d_{k^*+1}^2\bar{H}(k^*+1)} \tag{34}$$

Therefore, $\bar{h}$ can be written as

$$\bar{h} = (I) + (II). \tag{35}$$

We will give an upper bound of (I) and (II).

Using the definition of $\bar{H}(k^*-1)$ (see 2), we get

$$d_{k^*-1}^2\bar{H}(k^*-1) = d_{k^*-1}^2\sum_{k\in\{k^*-2,k^*\}}\frac{1}{(d_{k^*-1}+d_k)^2}$$

Since $d_{k^*} = 0$ and $d_{k^*-2} \geq d_{k^*-1}$, we get

$$d_{k^*-1}^2 \bar{H}(k^*-1) \leq 1 + \frac{1}{4} = \frac{5}{4} \tag{36}$$

By applying (36) in (33) we get

$$(I) \leq \frac{4}{5} \tag{37}$$

Using the definition of $\bar{H}(k^*+1)$ ((see 2)) we get

$$d_{k^*+1}^2 \bar{H}(k^*+1) = d_{k^*+1}^2 \sum_{k \in \{k^*, k^*+2\}} \frac{1}{(d_{k^*+1} + d_k)^2}$$

Since $d_{k^*} = 0$ and $d_{k^*+2} \geq d_{k^*+1}$, we get

$$d_{k^*+1}^2 \bar{H}(k^*+1) \leq 1 + \frac{1}{4} = \frac{5}{4}. \tag{38}$$

By applying (38) in (34) we get

$$(II) \leq \frac{4}{5} \tag{39}$$

Applying (37) and (39) in (35), we get

$$\bar{h} \geq \frac{4}{5} + \frac{4}{5} = \frac{8}{5}.$$

Putting the value of $\bar{h}$ in Corollary, we get the required bound as given in (4). □

## A.3 Proof of Theorem 2

*Proof.* We will upper bound the error probability as given by $P_{\boldsymbol{p(\mu)}}(\hat{k}_{L+1} \neq k^*)$. For simplicity of notation, we will drop $\boldsymbol{p(\mu)}$ and we refer to it as $P(\hat{k}_{L+1} \neq k^*)$. The FB-BAUB runs for $T$ horizon in $L+1$ number of phases that satisfies (6), where $L = \frac{\log_2 K/3}{\log_2 3/2}$ and outputs the arm $\hat{k}_{L+1}$. We will now upper bound the probability of error as,

$$P(\hat{k}_{L+1} \neq k^*) \leq \sum_{l=1}^{L+1} P(k^* \text{ elim. in } l). \tag{40}$$

The best arm is eliminated in phase $l$ in the following cases:

1. $k^* \in \{k^M, \ldots, k^A\}$, and $\hat{\mu}_{k^B}^l$ or $\hat{\mu}_{k^N}^l$ is greater than both $\hat{\mu}_{x^M}^l$ and $\hat{\mu}_{k^A}^l$

2. $k^* \in \{k^B, \ldots, k^N\}$, and $\hat{\mu}_{k^M}^l$ or $\hat{\mu}_{k^A}^l$ is greater than both $\hat{\mu}_{k^B}^l$ and $\hat{\mu}_{k^N}^l$

The two cases are illustrated in Fig. 6. From Remark 1, $k^*$ will not get eliminated if $k^* \in \{k^A, \ldots, k^B\}$. However, we will upper bound the error probability by assuming that $k^*$ will always fall in the above two cases. Notice that Case 1 and Case 2 are symmetrical. Hence, we can consider that $k^*$ will always fall in either one of the cases. Without loss of generality, we consider Case 1.

$$\begin{aligned}
P(k^* \text{ elim. in } l) &\leq P(\hat{\mu}_{k^B}^l > \hat{\mu}_{k^M}^l \text{ and } \hat{\mu}_{k^A}^l | k^* \in \{k^M, \ldots, k^A\}) \\
&+ P(\hat{\mu}_{k^N}^l > \hat{\mu}_{k^M}^l \text{ and } \hat{\mu}_{k^A}^l | k^* \in \{k^M, \ldots, k^A\}) \\
&\leq 2P(\hat{\mu}_{k^B}^l > \hat{\mu}_{k^M}^l \text{ and } \hat{\mu}_{k^A}^l | k^* \in \{k^M, \ldots, k^A\}),
\end{aligned} \tag{41}$$

where the last inequality is due to the fact that, for Case 1, $\mu_{k^B} \geq \mu_{k^N}$ by unimodality. Now for Case 1, $\mu_{k^A}$ is always greater than $\mu_{k^B}$, but $\mu_{k^M}$ may not be greater than $\mu_{k^B}$. Then, we can further upper bound (41) as

$$P(k^* \text{ elim. in } l) \leq 2P(\hat{\mu}_{k^B}^l > \hat{\mu}_{k^A}^l | k^* \in \{k^M, .., k^A\}). \tag{42}$$

We will now apply Hoeffding's inequality Lattimore & Szepesvári (2020) in (42) as stated in the following Lemma.

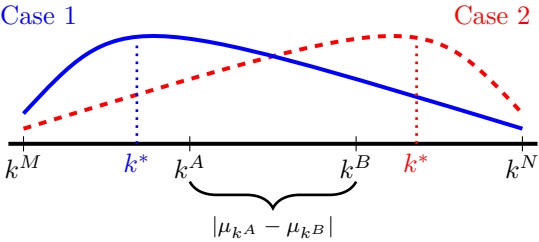

Figure 6: Different cases of elimination in any phase $l$. $k^*$ will not get eliminated if it is in between arms $k^A$ and $k^B$.

**Lemma 1** (Hoeffding's Inequality for Subgaussian Random Variables). *If $X_1, \ldots, X_m$ are $m$ i.i.d samples drawn from $\beta$-Subgaussian then for any $i \in [m]$, then*

$$P\left(X_i \geq \mu + \epsilon\right) \leq \exp\left(-\frac{\epsilon^2}{2\beta^2}\right) \ and \ P\left(\frac{1}{m}\sum_{i \in [m]} X_i \geq \mu + \epsilon\right) \leq \exp\left(-\frac{m\epsilon^2}{2\beta^2}\right).$$

Thereby, applying Lemma 1 in (42), we have

$$P(\hat{\mu}_{k^B}^l > \hat{\mu}_{k^A}^l) \leq \exp\left\{-\frac{1}{2}\frac{N_l}{4}\left(\frac{\Delta_{A,B}^l}{\beta}\right)^2\right\}, \tag{43}$$

where $\Delta_{A,B}^l = \mu_{k^A} - \mu_{k^B}$ for phase $l$ and is greater than 0 for Case 1. Using $\Delta$, which is defined as $\Delta = \min_{2 \leq i \leq K-1}\left|\mu_i - \mu_{i-1}\right|$, and the fact that there are at least $\frac{j_l}{3}$ arms between $k^A$ and $k^B$, for Case 1 we have, $\Delta_{A,B}^l \geq (j_l/3)\Delta$. Thus from (41) and (43) we have,

$$P(k^* \text{ elim. in } l) \leq 2\exp\left\{-\frac{N_l}{72}\left(j_l\frac{\Delta}{\beta}\right)^2\right\}. \tag{44}$$

Using $j_l = \left(\frac{2}{3}\right)^{l-1}K$ in (44), we can find the probability of the best arm getting eliminated in phases 1 and 2, phase $L+1$, and the rest of the phases separately. Using (8), we have

$$P(k^* \text{ elim. in } 1\&2) \leq 2\exp\left\{-\frac{TK}{32}\left(\frac{\Delta}{\beta}\right)^2\right\} + 2\exp\left\{-\frac{TK}{72}\left(\frac{\Delta}{\beta}\right)^2\right\}. \tag{45}$$

For phase $L+1$, since the best arm is selected among three arms when each arm is sampled $T/9$ times, we have

$$P(k^* \text{ elim. in phase } L+1) \leq 2\exp\left\{-\frac{T}{24}\left(\frac{\Delta}{\beta}\right)^2\right\}. \tag{46}$$

From (44), the error probability for the remaining phases is

$$P(\text{best arm elim. in phase 3 to phase L}) \leq 2\sum_{l=3}^{L}\exp\left\{-\frac{T}{8}\frac{K^2}{9}\left(\frac{2}{3}\right)^{2(l-1)}\frac{2^{L-l+1}}{3^{L-l+2}}\left(\frac{\Delta}{\beta}\right)^2\right\}$$

$$= 2\sum_{l=3}^{L}\exp\left\{-\frac{TK}{24}\left(\frac{2}{3}\right)^l\left(\frac{\Delta}{\beta}\right)^2\right\}$$

$$\leq 2(L-2)\exp\left\{-\frac{T}{8}\left(\frac{\Delta}{\beta}\right)^2\right\}. \tag{47}$$

By (40), (45), (46) and (47), we obtain the upper bound as

$$P(\hat{k}_{L+1} \neq k^*) \leq 2\exp\left\{-\frac{T}{24}\left(\frac{\Delta}{\beta}\right)^2\right\} + 2\exp\left\{-\frac{TK}{32}\left(\frac{\Delta}{\beta}\right)^2\right\}$$
$$+ 2\exp\left\{-\frac{TK}{72}\left(\frac{\Delta}{\beta}\right)^2\right\} + 2(L-2)\exp\left\{-\frac{T}{8}\left(\frac{\Delta}{\beta}\right)^2\right\}. \qquad \square$$

## A.4 The Empirical Number of Arm Pulls of the State-of-the-art Algorithms

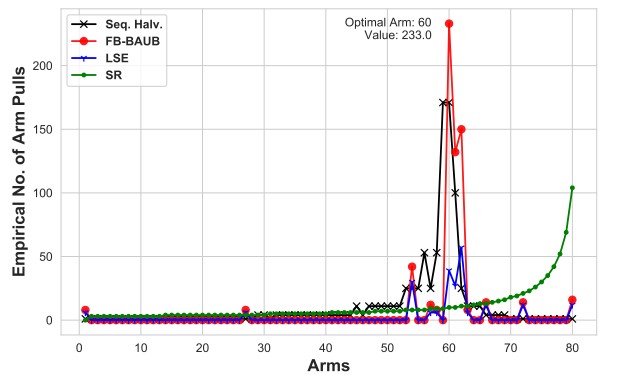

Figure 7: Expt. 1: No. of arm pulls for $K = 80$ arms for $T = 1000$.

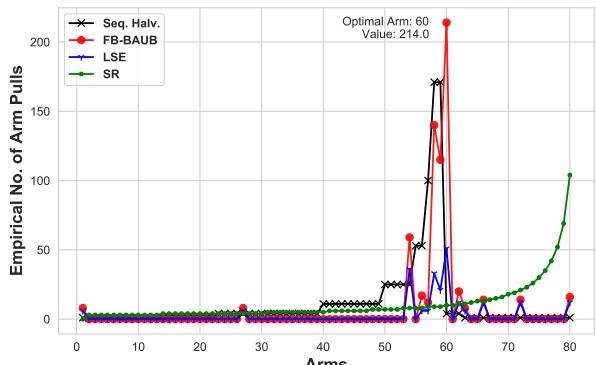

Figure 8: Expt. 2: No. of arm pulls for $K = 80$ arms for $T = 1000$.

Fig. 7 and Fig. 8 illustrate the empirical number of pulls of each of the $K$ arms, where $K - 80$, for Experiment 1 and Experiment 2 for $T = 1000$ budget, respectively. For both the experiments, the optimal arm is $k^* = 60$. Note that Exp. 2 is more challenging to identify the optimal arm compared to Exp. 1, as the means of the successive arms are not well-separated compared to that in Exp. 1. Therefore, for Exp. 2, the sub-optimal arms are explored more to distinguish them from the optimal arm. Therefore, a smaller budget will be available to explore the optimal arm for Exp. 2 compared to that of Exp. 1.

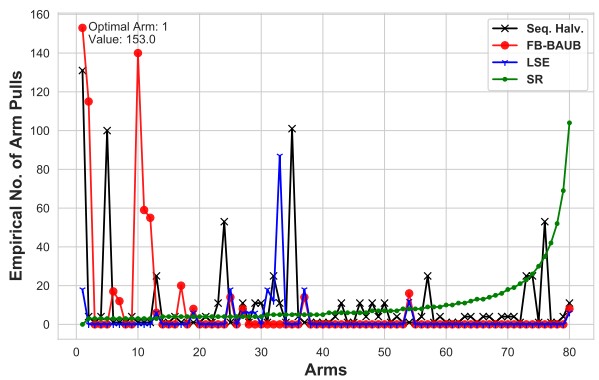

Figure 9: Expt. 3: No. of arm pulls for $K = 80$ arms for $T = 1000$.

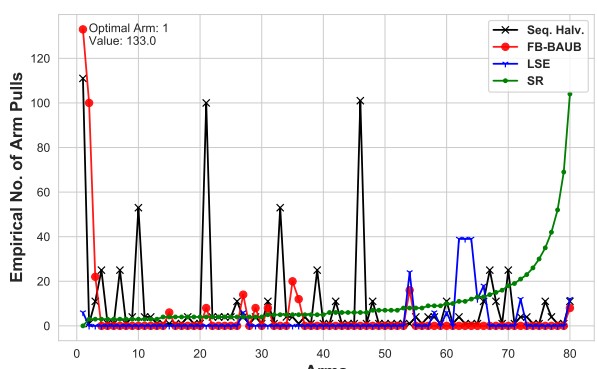

Figure 10: Expt. 4: No. of arm pulls for $K = 80$ arms for $T = 1000$.

Fig. 9 and Fig. 10 illustrate the empirical number of pulls of each of the $K = 80$ arms for Experiment 3 and Experiment 4 for $T = 1000$ budget, respectively. For both experiments, the optimal arm is $k^* = 1$. As Exp. 4 is more challenging to identify the optimal arm compared to Exp. 3, sub-optimal arms are explored more to distinguish them from the optimal arm. Therefore, a smaller budget remains to explore the optimal arm for Exp. 4 compared to that of Exp. 3.