# OpenReview forum: "Fixed Budget Best Arm Identification in Unimodal Bandits"
_TMLR — Accepted by TMLR_

### Review · Reviewer_fuwp · 2024-02-04

**Summary Of Contributions:**

This paper considers the best arm identification problem in a fixed budget stochastic multi-armed bandit setting, where the arm mean rewards exhibit a unimodal structure. The authors establish an error probability lower bound for misidentifying the optimal arm within a budget of $T$. In contrast to the lower bound for the unstructured case, the error exponent in this bound does not depend on the number of arms $K$ and is smaller by a factor $K \log K$, which captures the gain achievable by exploiting the unimodal structure. The authors also develop an algorithm named FB-BAUB that exploits unimodality to achieve the gain.

**Audience:**

Yes

**Broader Impact Concerns:**

I do not have ethical concerns.

**Claims And Evidence:**

Yes

**Requested Changes:**

Please see the review on weaknesses above.

**Strengths And Weaknesses:**

**Strengths:**

1.	The studied problem, best arm identification in unimodal bandits with fixed budget, is an interesting problem and finds real-world applications such as network throughput, sequential pricing, and bidding in online sponsored search auctions.
2.	The theoretical analysis in this paper is solid. This paper establishes an error probability lower bound for unimodal bandits with fixed budget, which shows the gain of utilizing the unimodal structure. In addition, this paper also designs an algorithm with error probability upper bounds.
3.	Empirical evaluations are provided to validate the effectiveness of the proposed algorithm.


**Weaknesses:**

1.	It would further improve this paper if the authors can discuss the gap between their upper bound and lower bound. The comparison to existing unimodal bandit or unstructured bandit results can help readers understand the contributions of this paper to the literature.
2.	The authors should discuss more on the novelty in analysis, compared to existing analytical procedure of unimodal bandits.

Overall, I think this submission is a good paper, and tend to accept.

---

> ### Author Response · Authors · 2024-02-12
> **Response to the comments of fuwp**
>
> Thank you for the comment. Our responses to these comments are grouped into two main key points, as follows:
>
> $\textbf{Discussion on the gap between their upper bound and lower bound of FB-BAUB:}$
> The upper bound of the error probability of FB-BAUB is of order $O(\log_2 K\exp(-T\Delta^2))$, and the lower bound of the error probability of fixed-budget BAI for unimodal bandits is of order $\exp(-T/\bar{H})$. We have acknowledged the gap between the upper and lower bounds for the fixed budget BAI for unimodal bandits, as the error exponent of FB-BAUB differs from the optimal error exponent with respect to the complexity terms as $\bar{H} \leq 2/\Delta^2$. Hence, FB-BAUB is not an optimal algorithm, and therefore, there is scope for improvement, representing a promising avenue for future work. However, our main focus is to improve the scaling of the error probability bound w.r.t. $K$. We have provided a near-optimal solution and have shown that the error exponent for both the lower and upper bounds does not involve $K$, quantifying the gain achieved by exploiting the unimodal structure. We are motivated by the study of unimodal bandits in the regret setting (Combes \& Proutiere (2014)), where the unimodal property helps improve the scaling of the regret bound with respect to $K$.
>
> To address this comment, we have revised the last paragraph of Sec. 6 of the revised paper (available at the anonymized link https://anonymous.4open.science/r/TMLR-Submission-6F64; the revised parts are highlighted in $\textbf{blue}$).
>
> $\textbf{Discussion on the novelty of the analysis of FB-BAUB:}$ Thank you for the comment. We have discussed the novelty of the analysis of FB-BAUB, compared to the existing analytical procedure for unimodal bandits, as follows:
>
> 1. $\textbf{Technical Challenges of FB-BAUB:}$ The PAC-bound provided by the LSE algorithm in Yu \& Mannor (2011) is based on the known $(\epsilon_l,\delta_l)$-PAC bound of the Sampling Algorithm (refer to Thm. 4.1) for every iteration. However, FB-BAUB does not run any sub-algorithm but has to carefully construct $N_l$ so that the budget constraint that balances the trade-off of elimination and exploration of new arms is attained. Note that $N_l$ are of different lengths over the phases, and the error bound of the FB-BAUB is obtained by carefully applying Hoeffding's inequality in each phase.
>
> 2. $\textbf{Minimum Gap Separation:}$ For a finite set of arms, LSE assumes that the gap between the mean rewards of the neighboring arms is separated by at least $D_L>0$ (Assumption 3.2 in Yu \& Mannor (2011)). More specifically, LSE requires knowledge of $D_L$, and its sample complexity is expressed in terms of $(\epsilon_l,\delta_l)$ for each phase $l$. However, FB-BAUB only considers that $\Delta >0$, i.e., the arm means to be distinct. We do not need any assumptions on the minimum separation of the mean rewards of the neighboring arms, i.e., FB-BAUB need not know $D_L.$ Therefore, FB-BAUB works when the arm means are distinct but arbitrarily close to each other, whereas LSE analysis assumes that this separation is at least $D_L$.
>
> 3. $\textbf{Analysis of fixed-budget setting vs. fixed-confidence setting:}$ Our analysis for the finite set of arms differs from that of Yu \& Mannor (2011), as we do not require the knowledge of $D_L$. Since $D_L$ is unknown, this is not the same as 'computing the sample size with a required estimation error'. We have to carefully control the elimination and decide the duration of the samples to meet the budget constraints. However, as rightly noted, in the case of Yu \& Mannor (2011), 'computing the sample size with a required estimation error' applies as they use the knowledge of $D_L$ for the finite set of arms.
>
> 4. $\textbf{Novelty in choice of $N_l$:}$ The nature of selecting an arm for sampling based on the golden ratio in Yu \& Mannor (2011) makes the separation between the arms selected for sampling non-uniform, which in turn makes the number of arms eliminated in each phase a random quantity. Hence, LSE is not a good strategy for fixed-budget settings where the number of samples is constrained, and we need to have a good accounting of the remaining arms for analytical traceability. On the other hand, in FB-BAUB, $2/3$ of the arms remain in each phase, and we increase the number of samples collected in subsequent phases by a factor of $3/2$. This helps us in two ways. (a) meet the budget constraint exactly across the phases, and (b) the number of samples in phase independent of the problem instance (like knowing $D_L$).
>
> To address this comment, we have added the above-mentioned points to the paragraph on ``Comparison with LSE'' of Sec. 6 and revised it in the revised version of this paper (available at the anonymized link https://anonymous.4open.science/r/TMLR-Submission-6F64; the revised parts are highlighted in $\textbf{blue}$).

---

### Review · Reviewer_GGpE · 2024-02-14

**Summary Of Contributions:**

This paper addresses the problem of Fixed budget best arm identification in multi-armed bandit problems where the expected rewards have a unimodal structure. It presents new error lower bounds for this problem, and proposes an algorithm with new error upper bounds. The effectiveness of the proposed algorithm is validated not only through theoretical analysis but also by numerical experiments.

**Audience:**

Yes

**Claims And Evidence:**

No

**Requested Changes:**

Regarding the weakness mentioned above, I would appreciate it if you could verify if there are misunderstandings in my review and respond with either a correction or a rebuttal.
Additionally, below are some minor points I have noticed:

* In the definition of $H_2$ in Section 3.3,
should the range of
$k$ be $k\neq 1$?
(from the definition of $\mu_{(k)}$, not $\mu_{(k^*)} = \mu_{k^*}$ but $\mu_{(1)} = \mu_{k^*}$)
* The usage of ordinal indicators such as "-th" or "-rd" appears to be inconsistent, with some instances using superscript formatting and others not.
* It seems correct to place footnotes after punctuation (see, e.g., Section 4.2 in the example PDF of https://neurips.cc/Conferences/2023/PaperInformation/StyleFiles).
* In the paragraph on Comparison with Sequential Halving in Section 6, it would be beneficial to add a reference to the literature proposing Sequential Halving. (I interpreted this as referring to the study by Karnin et al.(2013).)
* The statement "LSE eliminates about $1/\phi$ fraction of arms" may indeed be misleading. In LSE, $1/\phi$ appears to be the proportion of arms that are retained, not eliminated. The correct interpretation would be that the fraction of arms eliminated is $1-1/\phi$.

**Strengths And Weaknesses:**

Strengths:
* This paper is well-structured and easy to follow.
* The topic addressed in this paper appears to be of interest to the theoretical community of bandit algorithms.

Weaknesses:
* The comparison with Sequential halving in Section 6 raises some concerns.
The paper asserts that the bounds achieved by Sequential halving are $O(\log K \exp (-\frac{T \Delta^2}{K \log K}))$,
which seems to be derived from substituting $\frac{K}{\Delta^2}$ for $H_2$ in the bound of
$3 \log_2 K \cdot \exp(- \frac{T}{8 H_2 \log_2 K})$
presented in Theorem 4.2 by Karnin et al.(2013).
However, in Theorem 2 of the submission,
$\Delta=\min_{2\le i \le K} |\mu_i - \mu_{i-1}|$ is defined as the minimum difference in expected rewards between adjacent arms,
which implies $\Delta_i = \mu_{k^*} - \mu_{i} \ge |i-k^*| \Delta$,
and thus $H_2 \le \sup_{k > 1}\frac{k}{(\mu_{k^*} - \mu_{(k)})^2} \le \sup_{k > 1}\frac{k}{ (\Delta \lfloor k/2 \rfloor)^2} \le \frac{3}{\Delta^2}$.
Consequently,
the bounds achieved by Sequential halving would actually be
$3 \log_2 K \cdot \exp(- \frac{T \Delta^2}{24 \log_2 K})$.
From this observation,
the superiority of the bounds shown in Theorem 2 of this submission over those of Sequential halving --as the paper claims-- appears dubious.
I would welcome any corrections if my interpretation has any inaccuracies.
* The distinctions 1 and 4 claimed in Section 5.1 in comparison to LSE do not clearly emerge as fundamental.
The paper seems to advocate for replacing the division of intervals based on the golden ratio with a split into three equidistant intervals,
but the necessity and benefits of this approach remain unclear.
This difference appears to correspond to those between the golden ratio search (https://en.wikipedia.org/wiki/Golden-section_search) and ternary search (https://en.wikipedia.org/wiki/Ternary_search).
In typical optimization problems involving unimodal functions,
both the golden ratio search and ternary search can operate within a similar analytical framework,
yet it's well-known that the golden ratio search tends to be more efficient.
(In fact, the golden ratio search is mini-max optimal as shown by Kiefer, J. (1953), "Sequential minimax search for a maximum", Proceedings of the American Mathematical Society, 4 (3): 502–506)
Therefore,
the choice to opt for an approach akin to ternary search is surprising,
and I would be interested in hearing a compelling reason for this decision.

---

> ### Author Response · Authors · 2024-03-10
> **Response to the comments of GGpE**
>
> Thank you for the comment. Our responses to these comments are grouped into three main key points, as follows:
>
> $\textbf{Error bound of Sequential Halving in terms of $\Delta$:} $
> We feel the following inequality is not true in general, i.e., $\sup\limits_{k>1} \frac{k}{((k) - k^*)^2\Delta^2} \nleq \sup\limits_{k>1}\frac{k}{\lfloor k/2 \rfloor^2 \Delta^2}$, where $\Delta_{(k)} = \mu_{k^*} - \mu_{(k)} \geq |(k) - k^*|\Delta$. Below is a counter-example.
>
> Consider an example where $K=5, k^* =3$ and the mean of the arms are given as $\mu_1 < \mu_2 < \mu_3 > \mu_4 > \mu_5 \text{ and } \mu_5 < \mu_4 < \mu_1 < \mu_2.$ Therefore, the ordered arms are $(1):= 3, (2):= 2, (3):= 1, (4) = 4, (5) := 5$.
>
> The following table lists the values of $ \frac{k}{((k) - k^*)^2}$ and $\frac{k}{\lfloor k/2 \rfloor^2}$  for each $k$, as follows:
>
>  **$ k$**                                                | **$2$**                       | **$ 3$**                             | **$4$**                       | **$5$**
> :------------------------------------------------------------:|:-----------------------------:|:-----------------------------------:|:-----------------------------:|:-------------------------------:
>  **$\frac{k}{((k)-k^{*})^2}$**           | $2$               | $ \frac{3}{4}$                   | ${\bf 4}$         | $\frac{5}{4}$
>  **$\frac{k}{\lfloor k/2 \rfloor^2}$** | $2$ | $ {\bf 3}$ | $1$ | $ \frac{5}{4}$
>
> As seen, the values of $\sup\limits_{k>1} \frac{k}{((k) - k^*)^2}$ and $\sup\limits_{k>1}\frac{k}{\lfloor k/2 \rfloor^2}$ are not the same. Therefore, if there exists any $k \geq 3$ such that $(k) = k^*+1$, the inequality will not hold, i.e., $\sup\limits_{k>1}\frac{k}{((k)-k^*)^2} \nleq \sup\limits_{k>1}\frac{k}{(\lfloor k/2 \rfloor)^2}$.
>
> In our paper, we have considered the fact that $H_2 < H_1 =  \sum\limits_{k\neq k^*}\frac{1}{(\mu_{k^*} - \mu_k)^2}\leq \sum\limits_{k\neq k^*} \frac{1}{(k-k^*)^2\Delta^2}\leq\frac{K}{\Delta^2},$ as $\Delta_k = \mu_{k^*}-\mu_k \geq |k-k^*|\Delta \geq \Delta$. Using this inequality, we get the error bound of Sequential Halving as of order $O(\log K \exp(\frac{-T\Delta^2}{K\log K}))$. For better understanding, we revised the paragraph on ``Comparison with Sequential Halving'' in Sec. 6 in the revised paper (available at the anonymized link https://anonymous.4open.science/r/TMLR-Submission-6F64; the revised parts are highlighted in blue).
>
> $\textbf{Justification of not selecting arms by golden ratio ($\phi$):}$ In our method, $2/3$ of the arms remain in each phase, and we increase the number of samples collected in subsequent phases by a fraction of $3/2$. This helps us in two ways. (1) meet the budget constraint, and (2) the number of samples collected in phase independent of the problem instance (like knowing $D_L$). Moreover, if we use $1/\phi$ instead of $2/3$, it brings in at least two complications in the analysis, as highlighted below:
>
> 1. If we use $\phi$ in our method, the new phase lengths are $N'_l= (\frac{1}{\phi})^L(\phi)^{l-1}T,$ for $l= 1,2,\dots, L.$ Then, the total number of samples does not add up to $T$ for any $L$, i.e.,
>
> $\sum\limits_{l=1}^{L}N'_l = \frac{T}{\phi - 1}[1- (\frac{1}{\phi})^L] \neq T$.
>
> 2. Even if we fix some $L$ by some means (which we do not know how to do using $\phi$), we cannot deterministically know how many arms will remain when the budget is exhausted. This is because the separation between the arms chosen based on $\phi$ for sampling in each phase is non-uniform, and the number of arms getting eliminated can be different (unlike FB-BAUB, which is fixed). This makes the analysis challenging as we do not know how many arms are left at the end of the $L$-th phase.
>
> Hence, selecting arms using $\phi$ will be challenging for fixed-budget settings where the number of samples is constrained, and we need to have a good accounting of the remaining arms. Our strategy makes the analysis tractable with a clean bound without requiring the knowledge of $D_L$.
>
> $\textbf{Correction of the errors:}$ Thank you for your suggestions and for noticing the typos. We have made the following changes in the revised paper (available at the anonymized link https://anonymous.4open.science/r/TMLR-Submission-6F64; the revised parts are highlighted in $\textbf{blue}$):
>
> 1. We have redefined $H_2$ as $H_2 = \sup\limits_{k>1}\frac{k}{(\mu_{k^*} - \mu_{(k)})^2}$.
> 2. We have made consistency in using the ordinal indicators.
> 3. We have properly placed footnotes.
> 4. We cited Karnin et al. (2013) in the paragraph on ``Comparison with Sequential Halving''.
> 5. We changed the statement ''LSE eliminates about $1/\phi$ fraction of arms” to ''LSE eliminates about $1-1/\phi$ fraction of arms” in Sec. 5.1.
> 6. We have carefully proof-read the paper and have corrected all the typos we could find.

---

> > ### Comment · Reviewer_GGpE · 2024-03-30
> >
> > Thank you for your response and for revising the paper.
> > I still have some questions about the error bounds for Sequential Halving and would be happy to ask additional questions.
> >
> > > $\sup_{k>1} \frac{k}{((k)-k^*)^2 \Delta^2} \not\le \sup_{k>1} \frac{k}{( \lfloor k / 2 \rfloor )^2 \Delta^2}$
> >
> > I agree with the claim that the inequality $\sup_{k>1} \frac{k}{((k)-k^*)^2 \Delta^2} \le \sup_{k>1} \frac{k}{( \lfloor k / 2 \rfloor )^2 \Delta^2}$ does not hold in general.
> > In my earlier review comments, I did not claim that this inequality holds,
> > but that $\sup_{k > 1}\frac{k}{(\mu_{k^*} - \mu_{(k)})^2} \le \sup_{k > 1}\frac{k}{ (\Delta \lfloor k/2 \rfloor)^2}$ (I would appreciate it if you could check again just to be sure).
> > The example in your response is not a counterexample to the latter inequality as we have $(\mu_{k^*} - \mu_{(4)})^2 = (\mu_{3} - \mu_{4})^2 \ge (\mu_{3} - \mu_{1})^2 \ge 4 \Delta^2$,
> > where the first inequality follows from $\mu_4 < \mu_1 < \mu_2 < \mu_3$.
> >
> > In order to see that
> > $\sup_{k > 1}\frac{k}{(\mu_{k^*} - \mu_{(k)})^2} \le \sup_{k > 1}\frac{k}{ (\Delta \lfloor k/2 \rfloor)^2}$,
> > it suffices to show
> > $ (\mu_{k^*} - \mu_{(k)})^2 \ge \Delta \lfloor k/2 \rfloor $ for $k > 1$.
> > Under the assumption in Theorem 2 in the paper,
> > for any positive integer $i$,
> > the number of arms $k$ such that $\mu_{k^*} - \mu_{k} < \Delta \cdot i$ is at most $2i - 1$,
> > i.e.,
> > $|\\{ k \mid \mu_{k^*} - \mu_{k} < \Delta i \\} | \le 2 i - 1$.
> > This means that $\mu_{k^*}-\mu_{(2i)} \ge \Delta \cdot i$ for any positive integers $i$.
> > From this and the fact that $\mu_{k^*} - \mu_{k+1} \ge \mu_{k^*} - \mu_{k} $,
> > we have $\mu_{k^*}-\mu_{(k)} \ge \Delta \lfloor k/2 \rfloor$ for any $k>1$.
> >
> > > In our paper, we have considered the fact that $H_2 < H_1 = \sum_{k \neq k^*} \frac{1}{\mu_{k^*} - \mu_k}^2 \le \sum_{k \neq k^*} \frac{1}{(k-k^*)^2 \Delta^2} \le \frac{K}{\Delta^2}$,
> >
> > If we admit that $H_2 \le \sum_{k \neq k^*} \frac{1}{(k-k^*)^2 \Delta^2} $,
> > we can easily confirm that $H_2 = O( \frac{1}{\Delta^2} )$,
> > which is better than the bound of $O(\frac{K}{\Delta^2})$.
> > In fact,
> > we have
> > $ \sum_{k \neq k^*} \frac{1}{(k-k^*)^2 } \le \sum_{k=-\infty}^{k^*-1} \frac{1}{(k-k^*)^2} + \sum_{k=k^*+1}^{+ \infty} \frac{1}{(k-k^*)^2} = \sum_{k =1}^{+ \infty} \frac{2}{k^2} = \frac{\pi^2}{3} $ (solution to the Basel problem).
> > Even if the previous comment did not convince,
> > this consideration would confirm that Sequential Halving achieves a better bound.
> >
> > From the above, I am not yet convinced of the validity of the analysis for Sequential Halving.
> >
> > ---
> >
> > Minor comment:
> >
> > Theorem 4.2 by Karnin et al.(2013).
> > In my previous review comments, I wrote "Theorem 4.2 by Karnin et al.(2013)",
> > but this was a typo for "Theorem **4.1**".
> > This typo seems to have been carried over to the revision of your paper.
> > I apologize for the inconvenience,
> > but I would appreciate it if you could check and correct the citation.

---

> ### Author Response · Authors · 2024-04-01
> **Comparison with the Sequential Halving bound.**
>
> Thank you for highlighting the bound of Sequential Halving under the assumption that the minimum separation between the means of the neighboring arms is at least $\Delta>0$.
>
> We agree with your analysis. We have revised our claim that the improvement in error-bound by exploiting unimodality is of factor $\log K$ and removed the earlier claim that it is $K\log K$.
>
> The reference to Theorem 4.1  by Karnin et al.(2013) is also corrected.
>
> We have submitted the revised version with the changes highlighted in blue.

---

> > ### Comment · Reviewer_GGpE · 2024-04-05
> >
> > Thanks for the reply and the revision.
> > The authors' comments and revisions have now convinced me that the claims made in the submission are supported by accurate evidence.

---

> > > ### Author Response · Authors · 2024-04-08
> > > **Acknowledging review**
> > >
> > > Thanks for helping us correct the claims made in the paper. We will acknowledge the help in the paper!

---

### Review · Reviewer_NAro · 2024-02-26

**Summary Of Contributions:**

This study explores the problem of unimodal bandits within the context of fixed-budget best arm identification (BAI). The problem setting is characterized by an order in the mean rewards and a specific instance of the standard fixed-budget BAI. While the setting is restricted, the authors have established tighter lower and upper bounds alongside an optimal algorithm. Notably, $K$ is removed from the exponent in the probability of misidentification, marking a considerable theoretical advancement over strategies devised for more generalized settings.

**Audience:**

Yes

**Broader Impact Concerns:**

N / A

**Claims And Evidence:**

Yes

**Requested Changes:**

### Requests
- We should define the distribution $p_k(\mu_k)$ as a distribution which only depends on the mean parameters. For example, we do not consider various variances in the defined class. That is, we need to consider a class where only the means vary, but the other parameters are fixed.

### Suggestions
The following is not a request, but I just suggest the authors reconsider them.
- The authors repeatedly use the word "unimodal." However, it is unclear until the definition in Section 3. It may help the readers if the authors briefly introduce the setting in earlier parts, such as the Abstract or Introduction.
- The authors uses $p_k(\mu_k)$ to denote the distribution of a reward of arm $k$. I think that we do not have to use the subscript $k$ twice. Instead, we can use $p(\mu_k)$ to denote it. In fact, the authors defines $p_k(\mu_k)$ as $N(\mu_k, 1)$. Here, $N(\mu, 1)$ is fixed among arms, but the only mean parameters $\mu_k$ shift. In this sense, I considered that $p(\mu_k)$ is more appropriate to denote the distribution (but both are fine).
- Regarding the definition of the distributions, the authors mention that "samples from an unknown distribution $p_k(\mu_k)$." However, I think that the decision-maker may understand the distribution to some extent; the decision-maker does not know $\mu_k$ only. Therefore, "samples from a distribution $p_k(\mu_k)$ with an unknown parameter $\mu_k$" may be a more accurate expression.

**Strengths And Weaknesses:**

### Overall assessments.
The problem setting is a simple and straightforward extension of the standard BAI. However, it seems practical and useful. I also prefer the proposed approach since I have been engaged in dynamic pricing, which I think is a potential application of the proposed strategy.

From a theoretical viewpoint, less dependence on $K$ is interesting and matches our intuition.

The weakness of this study is a bit unrigorous mathematical arguments; however, it is still understandable. In particular, I worry about the definition of the bandit model. As I point out in the Requested Changes section, I think that the authors need to restrict the class more. For example, the authors should exclude a class of distributions such that both mean and variance shift.

In summary, while the idea and the strategy are straightforward and not so technical, the proposed algorithm is practical, and the theoretical findings are interesting. Although several parts should be fixed, I think that this study is worth publishing after revision.

### Questions.
- Could the authors give us some intuitive explanation as to why the $K$ disappears from the exponent of the upper bound? Specifically, I would like to know how it affects the behavior of strategies compared to strategies designed for the standard BAI setting. In general, the $K$ term in fixed-budget BAI appears from the behavior such that the decision-maker will draw each arm proportional to $T/K$ times. However, in an unimodal setting, the number of arms draws more concentration on some specific arms.
- (If the authors have time,) it might be interesting if the authors show the empirical number of arm draws in each arm in experiments.

---

> ### Author Response · Authors · 2024-03-10
> **Response to the comments of NAro**
>
> Thank you for the comment. Our responses to these comments are grouped into three main key points, as follows:
>
> $\textbf{Intuitive explanation about why K disappears from the exponent of the upper bound:}$
> Our algorithm runs in phases where four arms are selected in each phase irrespective of $K$. For each phase, we exploit the unimodal structure over these four arms and establish a tight bound on the error probability of eliminating the optimal arm region. This bound naturally depends on the number of arm plays and the gaps in the means of the arms in that phase. Using the minimum separation gap between the neighboring arms $\Delta$, we could bound this error for each phase $l>2$ as (see Eq (47))
>
> $\exp(-\frac{TK}{24}(\frac{2}{3})^l(\frac{\Delta}{\beta})^2) \leq \exp(-\frac{TK}{24}(\frac{2}{3})^L(\frac{\Delta}{\beta})^2) = \exp(-\frac{T}{24}(\frac{\Delta}{\beta})^2),$ where $L$ is the number of phases chosen appropriately.
>
> Note that in Sequential Halving, all remaining arms are played in each phase. In our case, we play only four of the remaining arms in each phase. This strategy, along with the unimodal property, enables us to derive an error exponent which does not scale with $K$.
>
> $\textbf{The empirical number of arm pulls of the state-of-the-art algorithms for each arm in experiments:}$
> Thank you for the comment. We have added Fig. 7, 8, 9 and 10, which show the behaviour of the state-of-the-art algorithms on the empirical number of arm pulls for each experiment, in Appendix A.4 of the revised paper. We observe that the empirical number of pulls of the optimal arm for our algorithm is higher than the state-of-the-art algorithms for each experiment. The revised paper is given at the anonymized link https://anonymous.4open.science/r/TMLR-Submission-6F64. The revised parts are highlighted in $\textbf{blue}$ in the revised version of the paper.
>
> $\textbf{Corrections in the revised paper:}$ Thank you for your suggestions. We have made the following changes in the revised paper (available at the anonymized link https://anonymous.4open.science/r/TMLR-Submission-6F64; the revised parts are highlighted in $\textbf{blue}$), as follows:
>
> (1) We have defined the unimodal structure in the third paragraph of the Introduction. However, for better clarification, we have defined the unimodal structure in the Abstract. We revised the first line of the Abstract in the revised paper, as follows:
>
> $\textit{We consider the best arm identification problem in a fixed budget stochastic multi-armed bandit in which arm means exhibit unimodal structure, i.e., there is only one local maximum.}$
>
> (2) We changed the notation of $p_k(\mu_k)$ to $p(\mu_k)$, and the revised the paper following this notation.
>
> (3) We have changed ''samples drawn from an unknown distribution $p_k(\mu_k)$" to ''samples drawn from a distribution $p(\mu_k)$ with an unknown mean $\mu_k$".
>
> (4) We have carefully proof-read the paper and have corrected all the typos we could find.

---

### Author Response · Authors · 2024-03-25

Dear Editor and Reviewers,

We have carefully reviewed and addressed all the comments provided by the reviewers. We kindly request the editor and the reviewers to revisit our responses and let us know if there are any concerns that need to be addressed. Thank you for your time and consideration.

---

### Decision · Action_Editor_PDyD · 2024-04-06

**Recommendation:** Accept as is

**Comment:**

This paper tackles the problem of best arm identification within a fixed budget setting for stochastic multi-armed bandit models exhibiting a unimodal structure. The authors first establish a lower bound of the probability of misidentifying the optimal arm, which captures the gain achievable by exploiting the unimodal structure. Then, they develop an algorithm named as Fixed Budget Best Arm Unimodal Bandits (FB-BAUB) to exploit unimodality and attain a tight upper bound. The paper is further enriched by comprehensive empirical evaluations demonstrating the superiority of FB-BAUB over state-of-the-art algorithms.

By establishing error probability lower bounds, the authors provide a deeper understanding of the fixed budget best arm identification problem in unimodal bandits. The FB-BAUB algorithm embodies a novel approach to utilizing unimodal properties, showcasing not only theoretical innovation but also practical applicability. Its parameter-free nature and ease of implementation further enhance the paper's appeal to both researchers and practitioners.

**Audience:**

Yes

**Claims And Evidence:**

Yes